# Performance Characterization of Low-cost Air Quality Sensors for Off-grid Deployment in Rural Malawi

Ashley S. Bittner[1], Eben S. Cross[2], David H. Hagan[2], Carl Malings[3], Eric Lipsky[4], and Andrew Grieshop[1]

[1]Department of Civil, Construction and Environmental Engineering, North Carolina State University, Raleigh, NC 27606, USA
[2]QuantAQ, Inc., Somerville, MA 02143, USA
[3]NASA Postdoctoral Program Fellow, Goddard Space Flight Center, Greenbelt, MD 20771, USA
[4]Department of Energy Engineering, Penn State Greater Allegheny University, McKeesport, PA 15132, USA

*Correspondence to*: Andrew Grieshop (apgriesh@ncsu.edu)

**Abstract.** Low-cost gas and particulate sensor packages offer a compact, lightweight, and easily transportable solution to address global gaps in air quality (AQ) observations. However, regions that would benefit most from widespread deployment of low-cost AQ monitors often lack the reference grade equipment required to reliably calibrate and validate them. In this study, we explore approaches to calibrating and validating three integrated sensor packages before a one year deployment to rural Malawi using collocation data collected at a regulatory site in North Carolina, USA. We compare the performance of five computational modelling approaches to calibrate the electrochemical gas sensors: k-Nearest Neighbor (kNN) hybrid, random forest (RF) hybrid, high-dimensional model representation (HDMR), multilinear regression (MLR), and quadratic regression (QR). For the CO, $O_x$, NO, and $NO_2$ sensors, we found that kNN hybrid models returned the highest coefficients of determination and lowest error metrics when validated. Hybrid models also were the most transferable approach when applied to deployment data collected in Malawi. We compared kNN-hybrid calibrated CO observations from two regions in Malawi to remote sensing data and found qualitative agreement in spatial and annual trends. However, ARISense monthly mean surface observations were 2 to 4 times higher than the remote sensing data, due to proximity to residential biomass combustion activity not resolved by satellite imaging. We also compared the performance of the integrated Alphasense OPC-N2 optical particle counter to a filter-corrected nephelometer using collocation data collected at one of our deployment sites in Malawi. We found the performance of the OPC-N2 varied widely with environmental conditions, with the worst performance associated with high relative humidity (RH > 70%) conditions and influence from emissions from nearby residential biomass combustion. We did not find obvious evidence of systematic sensor performance decay after the one year deployment to Malawi. Data recovery (30-80%) varied by sensor and season and was limited by insufficient power and access to resources at the remote deployment sites. Future low-cost sensor deployments to rural Sub-Saharan Africa would benefit from adaptable power systems, standardized sensor calibration methodologies, and increased regional regulatory grade monitoring infrastructure.

# 1 Introduction

Ambient air pollution is a leading cause of morbidity and premature mortality in Sub-Saharan Africa (SSA) (Murray et al., 2020). Air pollution in SSA is expected to increase over time given regional growth in population and energy demand combined with a biomass fuel dominated energy mix (Shikwambana and Tsoeleng, 2020; Stevens and Madani, 2016; Liousse et al., 2014; Amegah and Agyei-Mensah, 2017). However, regulatory air quality (AQ) monitoring is uncommon in many SSA countries, partially due to the high cost of reference grade equipment (Amegah, 2018; Petkova et al., 2013). Remote sensing is a valuable tool to address these data gaps, but satellite observations alone have various shortcomings relative to in situ measurements (Martin et al., 2019). Additional validation with reliable surface measurements is required, particularly in SSA (Malings et al., 2020). In the meantime, low-cost gas and particulate sensor packages provide an affordable, compact, and easily transportable approach to supplement air quality networks in regions where reference grade instrumentation is not accessible. Malawi, located in south-eastern Africa, provides a relevant context to investigate how low-cost sensors (LCS) can be used to address the global dearth of AQ observations. The Malawi Bureau of Standards published ambient air quality limits based on World Health Organization guidelines in 2005 (Mapoma and Xie, 2013; MBS, 2005), but there is no regulatory air quality monitoring program in the country to date. Previous studies of AQ in Malawi have primarily focused on indoor air quality or were unable to capture long-term trends (Fullerton et al., 2009, 2011; Jary et al., 2017; Mapoma and Xie, 2013). A dependable and affordable LCS monitoring network in Malawi could provide data to monitor the evolution of air quality and establish baselines for future AQ management.

Given the potential applications, LCS deployments are becoming common (Giordano et al., 2021). However, as the cost of LCS decreases, so may the selectivity, linearity, and accuracy. Electrochemical gas sensors are prone to interference and cross-sensitivities. Interference occurs when sensors respond to changes in temperature (T) and relative humidity (RH). Cross-sensitivities occur when sensors respond to the presence of gases other than the target analyte (Lewis et al., 2016; Mead et al., 2013). Failure to properly account for these during calibration can result in substantial measurement error under ambient conditions (Lewis et al., 2016; Cross et al., 2017; Castell et al., 2017; Mead et al., 2013). The calibration and application of LCS technologies to augment existing regulatory monitoring networks has been widely explored (Cross et al., 2017; Hagan et al., 2018; Malings et al., 2019a, b; Mead et al., 2013; Zimmerman et al., 2018; Li et al., 2021), but historically there has been little standardization in calibration approach or performance evaluation (Castell et al., 2017; Duvall et al., 2021; Morawska et al., 2018; Rai et al., 2017). In response to this, the U.S. Environmental Protection Agency (EPA) recently released two reports outlining testing protocols, metrics, and target values to evaluate the performance of ozone and fine particulate matter ($PM_{2.5}$) sensors for non-regulatory supplemental and informational monitoring applications in the U.S. (Duvall et al., 2021a, b). Unfortunately, there is no similar guidance for validating LCS for deployments in settings without in situ regulatory monitors. The deployment and evaluation of LCS packages in areas without existing AQ monitoring infrastructure is a growing research area (Chatzidiakou et al., 2019; Hagan et al., 2019; Subramanian et al., 2020,

2018). A lack of in situ regulatory monitors requires collocation, calibration, and validation at another site, potentially under a set of environmental conditions different from those of the target deployment environment. Advancements in laboratory chamber calibration may help resolve this issue. In a controlled environment, gas sensors can be exposed to and calibrated for a range of environmental conditions (i.e., gas concentration, RH, T, pressure, etc.), which may allow LCS cross-sensitivity and interference to be measured and controlled for before deployment (Williams et al., 2014b; Spinelle et al., 2016; Lewis et al., 2016; Spinelle et al., 2015). However, studies of low-cost particle sensors have observed better performance under laboratory versus field conditions (Rai et al., 2017). For example, previous long-term field assessments of the Alphasense OPC-N2 optical particle counter have observed large variability with changing seasons, environmental conditions, and background pollution levels (Bulot et al., 2019; Rai et al., 2017; Sousan et al., 2016). Low cost optical particle sensors can systematically overestimate mass concentrations under high RH (>70%) conditions due to hygroscopic growth of the particles (Crilley et al., 2018; Di Antonio et al., 2018), with errors ranging from 100 to 500% depending on aerosol hygroscopicity (Hagan and Kroll, 2020). Further, the complex chemical, physical, and optical properties of aerosol can complicate the field evaluation of low-cost particle sensors. For the Alphasense OPC-N2, particle composition may impact the sensor output by as much as a factor of 30 (Rai et al., 2017; Sousan et al., 2016). A recent modelling effort by Hagan and Kroll (2020) found that the optical properties and particle size distribution of the source aerosol can result in errors of up to 100% and 90%, respectively, in mass measurements made by low-cost optical particle sensors. Measurement errors were highest for strongly absorbing aerosol dominated by small (< 300 nm) particles. These traits can be characteristic of aerosol emitted by biomass-burning (Reid et al., 2005), a dominant source of ambient PM throughout SSA (Marais and Wiedinmyer, 2016; Queface et al., 2011; Liousse et al., 2014). Therefore, stringent quality assurance is necessary to ensure the validity of LCS particle measurements in this environment.

In this study, we calibrated and evaluated the "ARISense", a moderate-cost, integrated gas, particle, and meteorological sensor package (Aerodyne, Inc.) for long-term field deployment to Malawi. Our overarching goal was to assess the viability of augmenting and maintaining a small, temporary network of LCS monitors, until a more formal governmental regulatory monitoring system can be established. Given that comparison to regulatory grade equipment in Malawi was not possible, the objective of this work was to devise an alternative methodology to evaluate the ARISense technology (Section 2.1) for accuracy, precision, and stability over the 1-year pilot deployment. In Section 2.3 and 2.4, we describe collocations of the gas sensors (in North Carolina, USA) and particle sensor (in Mulanje, Malawi) with reference or semi-reference instruments (described in Section 2.2). We use collocation data and quantitative assessment metrics (described in Section 2.5) to compare the performance of five modelling approaches to calibrate the gas sensors (Section 3.1) and estimate error in the particle sensor data (Section 3.2). After deployment to Malawi (described in Section 2.6), we qualitatively assess how the ARISense performed in the field using contextual information about nearby emission sources, diurnal trends, and an inter-comparison of calibrated gas model observations (Section 3.3 and 3.4). In Section 3.5 and 3.6, we compare the deployment results to remote sensing and reanalysis data products and to surface measurements from similar environments in SSA. Finally, in

Section 3.7, we qualitatively assess the long-term stability of the sensor readings and calibration models in Malawi by comparing seasonally similar ambient data collected one year apart at the same location. In concluding (Section 4), we draw on these pilot results to characterize the benefits, limitations, and robustness of this technology and methodology for our application: collecting AQ data in under-studied and -resourced regions. Additionally, we offer guidance on considerations to improve future remote deployment efforts. Detailed analysis and discussion of more than three years of data collected in Malawi will be presented in a forthcoming complementary publication.

## 2 Methods

The ARISense were collocated with reference instruments in North Carolina (NC) before and after deployment to Malawi. One ARISense was collocated with a semi-reference PM instrument at a deployment site in Malawi to assess the performance of the integrated OPC-N2. Instrumentation, collocation, and calibration are covered in Sect. 2.1 – 2.4. Performance assessment metrics are given in Sect. 2.5. Calibrated ARISense were deployed to Malawi (Sect. 2.6) and compared to remote sensing data products (Sect. 2.7).

### 2.1 ARISense sensor packages

The ARISense package (Fig. S1) integrated the following sensors from Alphasense Ltd., UK: carbon monoxide (CO-B4), nitric oxide (NO-B4), nitrogen dioxide (NO2-B43F), total oxidants (Ox-B421), and the OPC-N2 optical particle counter. The ARISense reported voltage readings from electrochemical gas sensor working electrodes (WE) and auxiliary electrodes (AE). Sensor differential voltage ($\Delta V$) was calculated as WE – AE. The Alphasense OPC-N2 recorded counts in 16 size bins spanning particle diameters from 0.38 to 17.5 µm, meaning the OPC-N2 primarily measures coarse (> 2 µm) and some accumulation mode (0.1 to 2 µm) aerosols (Badura et al., 2018; Crilley et al., 2018; Sousan et al., 2016). Although the OPC-N2 has embedded algorithms to convert count measurements into mass concentrations of $PM_{1.0}$, $PM_{2.5}$ and $PM_{10}$ (particulate matter with aerodynamic diameters less than 1.0, 2.5, and 10 µm, respectively), the bin count data were manually integrated, converted to number concentration ($cm^{-3}$) assuming unity measurement efficiency across the bin range, and then to mass concentration assuming spherical particles with uniform density (1.65 g $cm^{-3}$). The values reported for $PM_{2.5}$ are $PM_2$. The location of the adjacent bin separations at 2.0 and 2.99 µm did not allow for direct estimates of $PM_{2.5}$. However, this was only one of many contributing sources of error in approximating true mass concentration with the Alphasense OPC-N2. Given the minimum cut-off diameter, we were unable to measure (nor did we try to estimate) the mass from particles smaller than 0.38 µm.

We used four ARISense monitors in this study: serial numbers ARI013, ARI014, ARI015 (Version 1.0, 2017), and ARI023 (Version 2.0, 2018). The monitors were powered by solar panels charging external batteries and recorded data to an internal USB device. Details and images are provided in Sect. 1 of Supplementary Information. Additional environmental and

meteorological sensors (i.e., T, RH, pressure, solar intensity, and noise) and system design are described in Cross et al. (2017).

## 2.2 Reference instrumentation

Gas concentration measurements for $NO_x/NO/NO_2$ (Teledyne Model T200UP), CO (Thermo Scientific Model 48i-TLE), and Ozone (Ecotech Federal Equivalent Method instrument) were obtained from reference instruments operated by the North Carolina Department of Environmental Quality (NC-DEQ) and the U.S. EPA.

The semi-reference MicroPEM (RTI International) instrument was used to assess the performance of the OPC-N2 in
Malawi. The MicroPEM, equipped with T and RH sensors, sampled (0.50 L/min, 100% duty cycle) via a $PM_{2.5}$ inlet into a nephelometer (0.1 Hz) and 25 mm PTFE filter. In previous evaluation studies, after gravimetric correction, the MicroPEM real-time nephelometer agreed with fixed-site reference monitors across a wide range of ambient PM concentrations (Du et al., 2019; Williams et al., 2014a). However, deployments observed baseline (zero) drift and poor performance at RH conditions above 94% (Williams et al., 2014a; Zhang et al., 2018). To account for baseline drift, the MicroPEM was zeroed
before each deployment using a HEPA filter. Additional details on the MicroPEM sensor, filter analysis, and quality assurance are provided in Sect. 1 of Supplementary Information.

## 2.3 Gas sensor collocation and calibration

Before deployment to Malawi, ARI013, ARI014, and ARI015 were collocated with EPA and NC-DEQ reference instruments (Fig. S2) at a near-highway site near Durham, North Carolina, USA (35.865°N, 78.820°W) between 29 May and
15 June 2017 (boreal summer – warm, mild season). ARI013 and ARI014 were collocated for 17 days. ARI015 was collocated for only 8 days due to a defect identified early in the collocation. All data were recorded at 1 minute resolution. Collocation site details are provided in Sect. 2 of Supplementary Information.

The pre-deployment collocation data were used to train, assess, and compare the performance of five modelling approaches
to convert the raw voltage data to concentration units and to account for sensor interference and cross-sensitivities. Outlying data points in the raw ARISense gas sensor voltage data due to noise and power cycling were visually identified and removed. Raw NO sensor data collected within 8 hours of a power cycle were also removed due to the extended warmup time of the NO-B4 sensor. ARISense data were time aligned with the reference data and both datasets were averaged to 5-min resolution. A random 70% of the collocation data were used for model training and the remaining 30% were withheld
for testing. Performance assessment metrics were calculated only for the withheld data.

Individual calibration models were built for each gas sensor ($O_x$, NO, $NO_2$, CO) in each monitor (ARI013, ARI014, ARI015) using five modelling approaches: k-Nearest Neighbor (kNN) hybrid (Hagan et al., 2018), Random Forest (RF) hybrid

(Malings et al., 2019a), High-Dimensional Model Representation (HDMR) (Cross et al., 2017), quadratic regression (QR) (Malings et al., 2019a), and multi-linear regression (MLR). The five models were selected for consideration based on their performance in previous studies. The kNN hybrid model was found to enable accurate measurements even when pollutant levels were higher than encountered during calibration (Hagan et al., 2018). Given that we expected levels of some pollutants to be higher in Malawi than during calibration in NC, we expected kNN hybrid models to be well suited for our application. Further, the kNN hybrid approach is expected to be widely applicable to a range of pollutants, sensors, and environments (Hagan et al., 2018). In a calibration and validation study conducted by Malings et al. (2019), RF hybrid models were recommended for any low-cost monitor using electrochemical sensors similar to their sensor package, the Real-time Affordable Multi-Pollutant (RAMP) monitor. Given that the RAMP and ARISense monitors use the same electrochemical sensors and have similar integrated designs, we expected RF hybrid models to perform well for our dataset. HDMR models were found to effectively model interference effects derived from the variable ambient gas concentration mix and changing environmental conditions over three seasons for the sensor types used in the ARISense package (Cross et al., 2017). Finally, MLR and QR are simple, popular calibration approaches and they were included in this study for that reason.

**Table 1:** Calibration modelling inputs for each gas sensor (CO = carbon monoxide, NO = nitrogen oxide, $NO_2$ = nitrogen dioxide, $O_x$ = oxidants) and model combination ('All' indicates k-nearest neighbor (kNN) hybrid, random forest (RF) hybrid, high-dimensional model representation (HDMR), multi-linear regression (MLR), and quadratic regression (QR). $\Delta V$ is the voltage difference between the working electrode (WE) voltage and the auxiliary electrode (AE) voltage measured by each electrochemical gas sensor, RH = relative humidity, T = temperature, DP = dew point.

| Gas Sensor | Data Inputs to Model | Models applied |
|---|---|---|
| CO | CO $\Delta V$, RH, T, & DP | All |
| NO | NO $\Delta V$, RH, T, DP, & NO WE[a] | All except QR |
| $NO_2$ | $NO_2$ $\Delta V$, RH, T, & DP | All except QR |
| $O_x$ | $O_x$ $\Delta V$, DP, & NO2 $\Delta V$ [b] | All except QR |

[a]kNN hybrid only

[b]RF hybrid only

The modelling inputs are summarized in Table 1. $O_3$ models were designed to account for sensor cross-sensitivity to $NO_2$ (Cross et al., 2017). Note that references to '$O_3$' indicate estimates made from calibrating the $O_x$ sensor data. References to '$O_x$' indicate raw voltage measurements from the total oxidant sensor. 'Ozone' is used when referring to the gaseous air pollutant. For our study, the CO HDMR models were set to allow only first-dimensional interactions, as second-order interactions were observed to lead to spurious results for data collected outside the bounds of training data (see Sect. 3.3 - on

deployment conditions). For the CO sensors, this effectively made the HDMR model equivalent to the MLR model. Therefore, the statistical metrics achieved by both models were identical and are shown as overlaid points on Fig. 2a.

## 2.4 OPC-N2 collocation and calibration

ARI023 was collocated with a MicroPEM in an ambient, combustion source-influenced environment on a house rooftop (4 m above ground level) in Mikundi village in Mulanje District, Malawi (16.056°S, 35.535°E) between 25 July 2018 and 7 August 2018 (austral winter – cool, dry season). We collected 130 hours of collocation data over three multi-day collection periods (i.e., 3 PTFE filters). A 75% completeness requirement was applied before the raw 1 min data were averaged to 1 h and 24 h intervals. Sub-daily averaging intervals were used to assess the OPC-N2 for near real-time (1 min) and diurnal trend (1 h) monitoring applications. A bin-wise RH-correction algorithm based on κ-Köhler theory was applied to correct for hygroscopic growth under high RH conditions, initially assuming particle density (ρ) equal to 1.65 g cm$^{-3}$ and aerosol hygroscopicity (κ) of 0.6 (Di Antonio et al., 2018). To observe sensitivity of this correction to the assumed hygroscopicity, the density was held constant at 1.65 g cm$^{-3}$ and the κ value was varied (κ = 0.15, 0.6, and 1). To observe variability due to the assumed source of the aerosol, the density and hygroscopicity were varied to approximate ammonium nitrate, dust, wildfire, and background aerosols. Aerosol property assumptions (κ and density) are based on Hagan and Kroll (2020) & Petters and Kreidenweis (2007).

## 2.5 Assessment metrics

We adapted performance metrics and target values from recently published U.S. EPA guidelines (Duvall et al., 2021a, b) to assess ARISense performance (Table S1). The EPA guidelines suggest using linearity, bias, precision, and error metrics to assess air sensor performance and they offer target values for each. We use the U.S. EPA target values as a quantitative marker to indicate satisfactory or unsatisfactory sensor performance, however given the differences in our study compared to the U.S. EPA methodology, we do not consider these categorizations to be definitive. Further, we emphasize that even if a sensor meets, or surpasses, the performance target values for each metric, this does not constitute endorsement by the U.S. EPA. Their guidelines were developed for $O_x$ and $PM_{2.5}$ air sensors, and we used these to assess the ARISense Ox-B421 and OPC-N2 sensors, respectively. Although there are no formal guidelines for CO, NO, and $NO_2$ sensors at the time of writing, for coherency, we opt to assess those sensors using a similar approach.

The coefficient of determination ($R^2$), an indicator of the correlation between estimated and true concentrations, was used to assess linearity. The root mean square error (RMSE) was used to assess error in the estimated measurements compared to the true values. The coefficient of variation (cV) was used to assess precision. Finally, to assess bias, a linear regression model (y = mx + b) was fit using the ARISense measurements as the dependent variable (y) and the reference measurements as the input variable (x), and the resulting slope (*m*) and intercept (*b*) were calculated. Quantitative descriptions for each metric are given in Sect. 3 of the Supplementary Information.

In addition, prediction intervals between the OPC-N2 and MicroPEM data were calculated to provide a statistical confidence interval to interpret OPC-N2 sensor measurements collected after the evaluation period (Bean, 2021). We calculated 68% (1-sigma) prediction intervals for the ARISense using collocation data from ARI023 (Table 2) collected at the Village 2 site (Fig. 1d). The 60-min averaged observations were used to fit a linear model, which required a Box-Cox transformation (Box and Cox, 1964) to obtain normally distributed residuals (Fig. S3). Details are given in Sect. 3 of the Supplementary Information.

## 2.6 Deployment to Malawi

ARI013, ARI014, and ARI015 were deployed to their respective monitoring locations in Malawi from July 2017 to July 2018 (shown as blue markers on Fig. 1). The three locations were selected to provide measures of regional variation and replicates in two paired village sites. ARI013 ("Village 2" site) and ARI014 ("Village 1" site) were deployed < 5 km apart (Fig. S5) in two rural villages in Mulanje, Malawi, adjacent to private residences. ARI015 ("University" site) was deployed >375 km northwest of the village sites at a rural university campus ~30 km from the capital city (Fig. S6). Additional satellite images are given in Sect. 4 of Supplementary Information.

Almost all rural households in Malawi (99.7%) use solid fuels (e.g., firewood, charcoal) for cooking (National Statistics Office, 2017). Emissions from widespread biomass cookstove use are known to impact local ambient air quality (Aung et al., 2016; Zhou et al., 2011; Amegah and Agyei-Mensah, 2017). Homes regularly using biomass cookstoves within 50 m of the monitoring sites were visually identified at the onset of the study (shown with red 'X's on Fig. 1c-d).

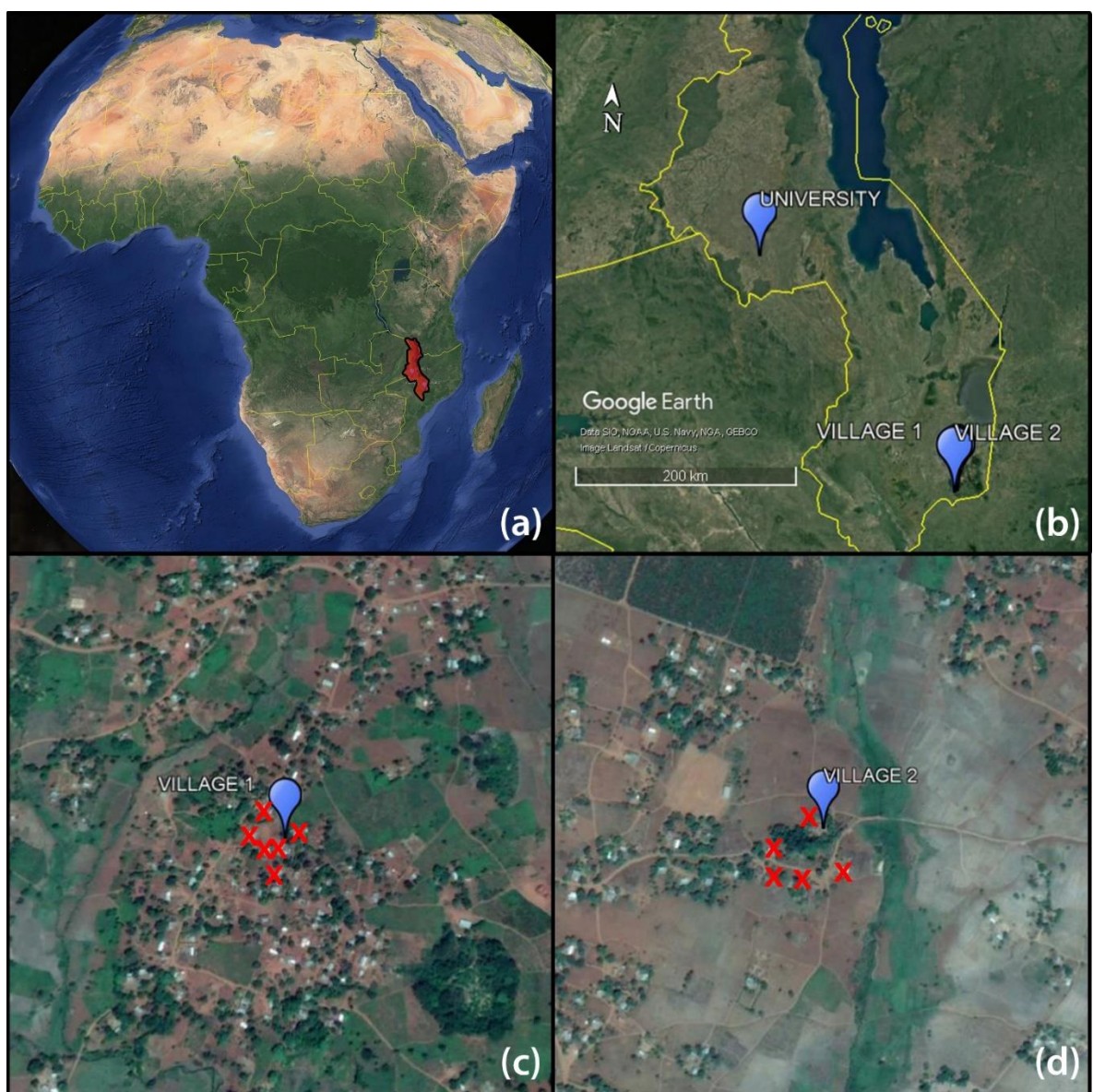

**Figure 1:** (a) Satellite map of Malawi in southeast Africa, (b) three ARISense monitoring sites in Malawi, (c) satellite map of Village 1, and (d) satellite map of Village 2. Blue markers indicate ARISense monitoring sites. Red 'X's indicate the location of known biomass cookstoves within 50 m of the monitoring site. Image source: Google Earth Pro Version 7.3.4.8248. *University, Village 1, and Village 2, Malawi, South-eastern Africa.* Borders and labels layer. Accessed: June 5, 2020. © Google Earth 2021.

A timeline of the ARISense collocations and deployments is given in Table 2. After the one year ambient deployment was completed, the ARISense were used for high-concentration emissions monitoring experiments in rural Malawi in July and August 2018. The details of those experiments (i.e., number of experiments, duration, approximate CO concentrations) are discussed in Sect. 5 of the Supplementary Information. We explore the impact of these experiments on sensor operation, but we do not discuss the data itself in this paper.

**Table 2**: Project timeline of collocations, deployment, and emissions monitoring experiments. The description under each period indicates the activity conducted during that timeframe. The location of the activity is given in parenthesis.

| ARISense | May - June 2017 | July 2017 - July 2018 | July - Aug 2018 | Aug 2018 - Mar 2019 |
|---|---|---|---|---|
| ARI013 | Collocation (NC) | Deployment (Village 2) | Emissions monitoring (Village 2)[a] | Collocation (NC) |
| ARI014 | Collocation (NC) | Deployment (Village 1) | Emissions monitoring (Village 2)[a] | Collocation (NC) |
| ARI015 | Collocation (NC) | Deployment (University) | Emissions monitoring (Village 2)[a] | n/a |
| ARI023 | n/a | n/a | OPC-N2 collocation (Village 2) | n/a |

[a]Data from emissions monitoring experiments not discussed in this paper

At the conclusion of the emissions monitoring experiments, ARI013 and ARI014 were returned to NC and were collocated with reference instruments at the near-highway Durham, NC site used in the pre-deployment collocation (described in Sect. 2.2). ARI015 was relocated to a new monitoring site in Malawi.

**2.7 Remote sensing and reanalysis data**

Two publicly available NASA data products were obtained from the Goddard Earth Sciences Data and Information Services Center (GES-DISC) Interactive Online Visualization and Analysis Infrastructure (GIOVANNI): 1) area-averaged, monthly Multispectral CO Surface Mixing Ratio (Daytime/Descending) from MOPITT and 2) CO Surface Concentration - ENSEMBLE from MERRA-2, henceforth referred to as "MOPITT" and "MERRA-2", respectively. MOPITT is a calibrated satellite observation and MERRA-2 is a global reanalysis data product. MERRA-2 is the output of an atmospheric chemistry

model that has assimilated other data, including satellite data, in making its estimations. Monthly averaged MOPITT and MERRA-2 observations were compared to ARISense CO surface data collected at the Village and University locations. Given the physical proximity of Village 1 and Village 2, and the similarity in monthly mean CO concentration at each site (Fig. S7), the average of the data sets ("Village Mean") was used. Additional details are given in Sect. 6 of the Supplementary Information.

## 3 Results and discussion

### 3.1 Gas sensor performance during collocation

Raw gas sensor voltages (5-min averaged data) from all three ARISense monitors (ARI013, ARI014, ARI015), excluding the $O_x$ sensor in ARI015, were highly correlated ($R^2 > 0.8$) during the pre-deployment collocation, suggesting changes in sensor response were due to environmental changes, not sensor-to-sensor variability (Fig. S9). The sensors in ARI013 and ARI014 were most closely correlated ($R^2 > 0.9$). The raw ARI015 $O_x$ sensor data showed weaker temperature dependence and the lowest correlation ($R^2 < 0.6$) with $O_x$ sensors in ARI013 and ARI014 (Fig. S9).

Figure 2 shows two performance metrics representing each sensor-model combination for the three ARISense. Data points toward the lower left corner of each Fig. 2 panel indicate better performance. Results from all ARISense-sensor-model combinations for all five performance metrics are given in Tables S4-S6. We found that performance varied by ARISense monitor, but none of the ARISense consistently performed better than the others. Overall performance varied by gas sensor type and modelling approach. The calibrated $NO_2$ sensors in all three ARISense were the least correlated with reference measurements compared to the other gas sensors. Only the ARI015 $NO_2$ sensor, calibrated by the RF hybrid model, surpassed the target value for the linearity metric ($R^2 > 0.8$). Further, no $NO_2$ sensor-model combination met the bias target values for slope and intercept. For all three ARISense, the calibrated $NO_2$ sensors underestimated the true concentration compared to the reference ($0.26 < m < 0.71$). However, all $NO_2$ sensor-model combinations met the error target (RMSE < 5 ppb) and approached the precision metric target.

At the other end of the performance spectrum, the calibrated $O_3$ sensors performed the best compared to the other gas sensors during pre-collocation. Nearly all $O_3$ sensor-model approaches attained similar linearity and error metrics ($0.85 < R^2 < 0.99$ and $2 < RMSE < 5$ ppb), well within the target values. Only the ARI015 $O_x$ sensor calibrated by the RF model failed to meet the RMSE target value, yet it returned the highest $R^2$ value compared to the other models. Additionally, all $O_x$ sensor-model combinations met the slope and intercept target values for bias. For the kNN hybrid model, the calibrated $O_3$ observations had a slope approximating 1 ($m > 0.98$) and an intercept of 0, suggesting minimal bias. Only the precision values ($37\% < cV < 54\%$) were outside the EPA guideline target range ($cV < 30\%$).

Most NO sensor-model combinations met the target value for the bias, error, and linearity metrics, but precision was low for all combinations assessed, with most cV values > 100%. This suggests that the variation in the NO data set was due to the raw sensor or reference measurements, rather than the modelling approaches. The MLR model was associated with the worst performance for all three NO sensors compared to the other models. However, for ARI015, all NO sensor-model combinations surpassed the target for every metric except precision. Again, the ARI015 gas sensor-RF hybrid model combination was the outlier compared to ARI013 and ARI014 sensor-model combinations (Table S6). We hypothesize that

the shorter collocation period of ARI015 (8 days compared to 17 days of collocation for ARI013 and ARI014) led some of the sensor-model combinations to be overfit or poorly constrained.

Most CO sensor-model combinations met or approached the target values for bias, linearity, and precision. The U.S. EPA recommended $O_x$ target values for these three indicators (Table S1) can be used to compare against the CO sensor values to approximate performance, but we surmise the error target value (RMSE $\leq$ 5 ppb) cannot. The U.S. EPA National Ambient Air Quality Standards suggest CO concentrations are 1-2 orders of magnitude larger than ambient ozone or $NO_x$

concentrations. By extension, we posit that a reasonable error target value for the CO sensor is 50 ppb. Except for the CO-kNN hybrid model combination, most CO sensor-model combinations did not meet our adapted error target value. However, considering the magnitude differences, the CO sensor-model combinations performed similarly to the NO, $NO_2$ and $O_x$ sensors in terms of error. The CO RMSE values (40-70 ppb) were correspondingly one order of magnitude larger than NO, $NO_2$, and $O_3$ RMSE values (2-7 ppb).

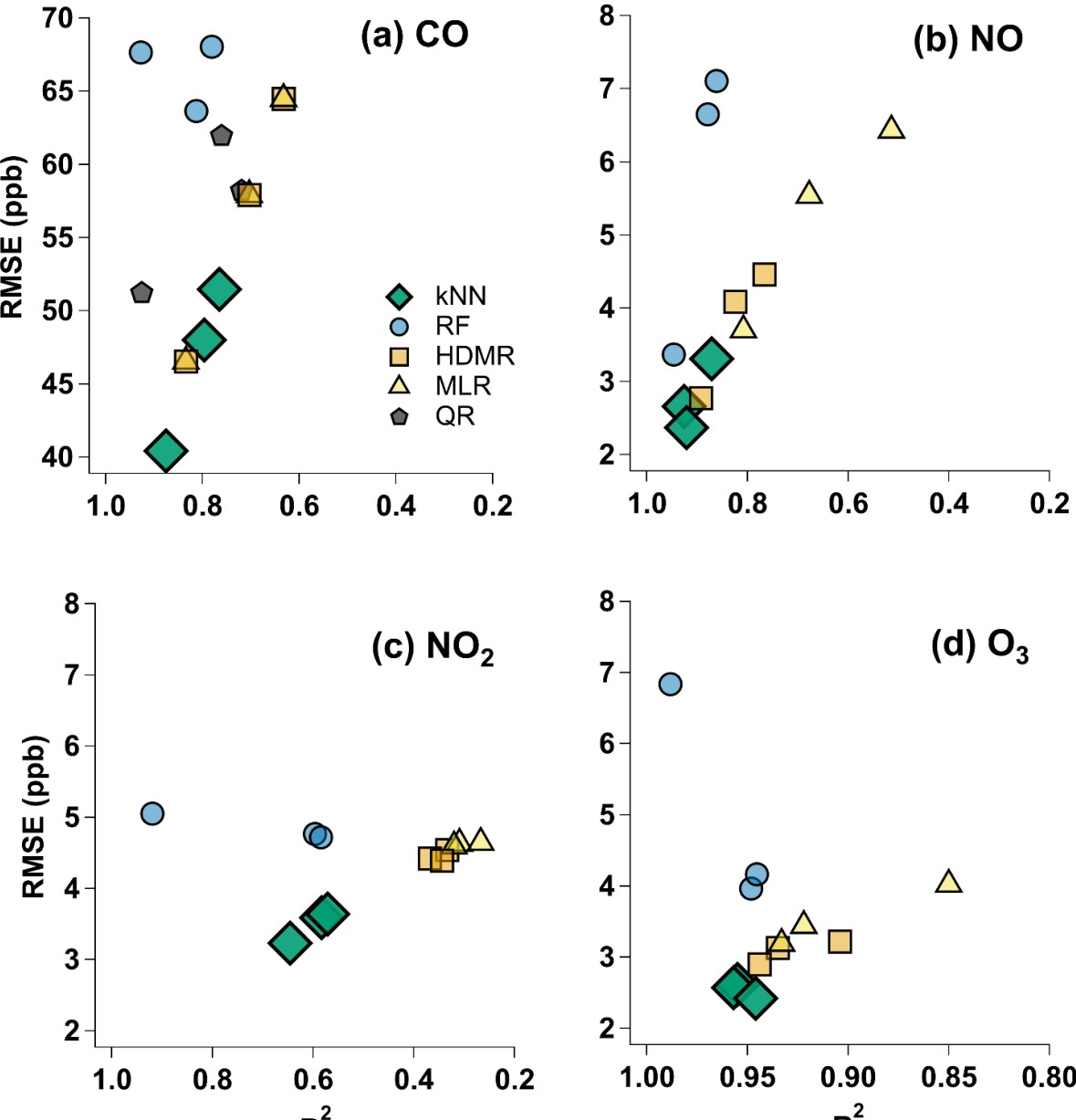

**Figure 2:** Performance comparison of gas sensors (a) CO, (b) NO, (c) $NO_2$, and (d) $O_3$ as calibrated by the five types of modelling approaches adopted for this study (kNN hybrid, RF hybrid, HDMR, MLR, QR). The model type is indicated by color and marker shape. An individual data point represents the paired metrics (RMSE and $R^2$) for one ARISense monitor. Since there are three ARISense (ARI013, ARI014, ARI015) monitors, there are three markers for each gas sensor-model combination. RMSE is root mean square error. $R^2$ is the coefficient of determination (-infinity $\leq R^2 \leq 1$). The lower left corner region of each panel indicates the highest performance based on these metrics.

For the suite of gas sensors in the ARISense monitors, we found the kNN hybrid model to be the best among the modelling approaches used in the pre-deployment collocation testing (Fig. 2). In almost all cases, the kNN hybrid model returned higher $R^2$ values, slope values closer to one, and lower RMSE values than any other model. The RF hybrid model attained similar, and occasionally higher $R^2$ values than the kNN hybrid, but it had higher (and therefore worse) RMSE values by comparison. Further, the kNN hybrid model showed the least inter-monitor variation in performance. In Fig. 2b-d, the kNN hybrid points are closely clustered together, suggesting that this model was able to attain similar performance for each of the three ARISense. Conversely, the other models, in particular the RF hybrid and MLR, showed a wide range in performance across the three ARISense. Even if another model was able to attain performance metrics higher than the kNN hybrid (e.g., HDMR and MLR CO models in Fig. 2a) it was only for one of the three ARISense monitors, never all three. Additionally, the MLR failed to meet target values for some ARISense-gas sensor combinations (Fig. 2a-b). Taken together, these findings suggest the kNN hybrid model is the best choice among these five modelling approaches for our application, given that we sought an approach uniformly applicable to all the gas sensors and all three ARISense.

### 3.2 OPC-N2 performance during collocation

Pre-deployment collocation $PM_{2.5}$ measurements in North Carolina (where no reference monitor/data were available) from ARI013, ARI014, and ARI015 suggest the Alphasense OPC-N2 sensors in each monitor responded similarly ($R^2 > 0.9$) when in the same environment (Fig. S10). ARI013 $PM_{2.5}$ mass concentration measurements were higher than measurements made by ARI014 and ARI015 (slope > 1), despite all ARISense being in the same location. ARI015 underestimated the mass at low concentrations compared to ARI013 and ARI014 (non-linear clustering at concentrations < 5 µg m$^{-3}$ in Fig. S10a and c). The OPC-N2 sensors in ARI014 and ARI015 showed the highest similarity (slope = $1 \pm 0.05$, $R^2 = 0.96$).

Figure 3 shows scatter plots of the ARI023 OPC-N2 and MicroPEM data collected during collocation at the Village 2 site in Malawi (Fig. S11). RH-correction partially mitigated the impact of overestimation due to hygroscopic growth but did not remove the artifact entirely (Fig. S12). RH-correction improved the precision and error metrics, bringing RMSE within the target value ($\leq 7$ µg m$^{-3}$) for the 24 h averaged data (Table S7). Increased averaging interval had a similar effect, but alone was insufficient to bring RMSE within the target range. Linearity was well below the target value ($R^2 > 0.7$) for all averaging intervals and RH-correction did little improve performance for this metric. For this data set, changes in bias and linearity appeared driven by averaging interval. For example, the OPC-N2 RH-corrected 1 minute data met the target for slope and intercept, but the 1 h and 24 h averaged data met neither. Particularly for the 24 h averaged data, the small sample was leveraged by a few points which drove metric values (Fig. 3c), however, close 1:1 agreement between the instruments was observed for four of the seven days of collocation. These results highlight the value of longer and more representative collocations. At least two 30-day collocations would be needed, during the hot-dry (Sep to Oct) and warm-wet (Nov to Apr) seasons, to characterize this specific site.

Even after RH-correction, the OPC-N2 overestimated mass concentrations compared to the nephelometer when RH was ≥ 70%. Conversely, the OPC-N2 often underestimated mass when RH was ≤ 30%. These effects were most noticeable at higher time resolutions (Fig. 3a-b). The effects of RH were tempered by a longer averaging interval, however for a particularly humid day at this site, the 24 h mass concentration was overestimated by a factor of three (Fig. 3c). Notably, the moderate-RH outliers in the 24 h average scatter plot suggest that other factors, in addition to RH, were contributing to error

in the OPC-N2 observations.

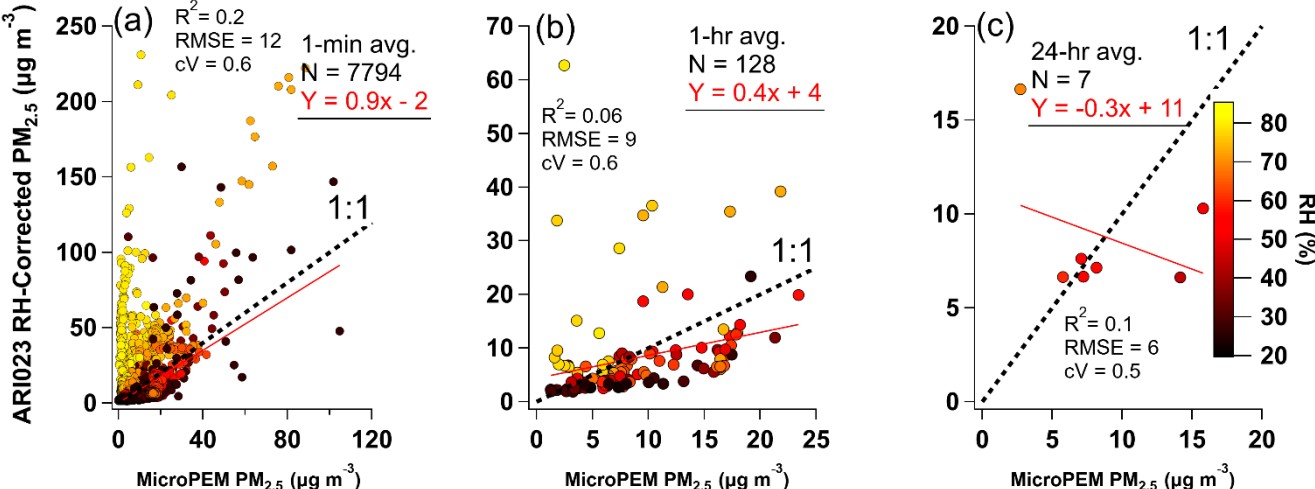

**Figure 3:** Scatter plots of RH-corrected $PM_{2.5}$ mass concentration measurements from the OPC-N2 versus mass-corrected
$PM_{2.5}$ measurements from the MicroPEM at 1 min (a), 1 h (b), and 24 h (c) averaging intervals. Data points are colored according to RH (%) conditions. The number of data points (N) and linear fit lines and regression coefficients (*m, b*) are given in red as Y = *m*x + *b*. Additional metric values are inset: $R^2$ is the coefficient of determination, RMSE is root mean square error (units of μg m$^{-3}$) assuming the MicroPEM is the reference instrument, and cV is coefficient of variation. The black, dashed line is a 1:1 line.

    To explore other contributors to variable OPC-N2 performance, Fig. 4 shows performance for RH-corrected data stratified by environmental conditions (wind direction, ambient concentration, and RH). Wind direction and concentration (Fig. 4a-b were selected to explore the possible effect of nearby cookstove emissions, while Fig. 4c highlights the remaining effect of
RH even after correction. We hypothesized that ambient concentration and wind direction might impact OPC-N2 performance given that the site was periodically exposed to cookstove emissions from the Village 2 site household kitchen (within 15 m to NW) and from adjacent residences (within 50 m to the S-SW in Fig. 1d). Figure 4 shows that wind direction was associated with performance variation, although to a lesser degree than RH. Slightly increased performance was

observed for northerly winds. Nearby cookstove use potentially explained the decreased performance associated with southerly winds. Four of the five morning cooking periods observed in the time series data were associated with wind blowing from the SE-S-SW (Fig. S14). Figure 4b shows that ambient concentration had a modest impact on OPC-N2 performance metrics. Linearity was expected to increase with concentration, particularly given that the high-concentration bin (20-105 µg m$^{-3}$) spanned a larger interval than the other bins. Precision within each concentration bin was low. The cV values were well beyond the recommended target value (cV < 30%). The OPC-N2 frequently underestimated the ambient mass concentration compared to the MicroPEM, particularly during higher concentration periods dominated by near-field biomass burning (i.e., slope = 0.4 for measurements between 20 to 105 µg m$^{-3}$). During periods of cookstove influence, the size distribution, hygroscopicity, and optical properties of the measured aerosol were likely altered. Assumptions about the source aerosol (density and hygroscopicity) used in the RH-correction were found to affect inferred OPC-N2 performance compared to the MicroPEM, though not predictably. For example, higher linearity and lower RMSE were observed when the particle composition was assumed to be highly hygroscopic ($\kappa = 1$), yet the least bias was observed at the lowest hygroscopicity assessed ($\kappa = 0.15$). Further, when the aerosol was assumed to be characteristic of wildfire (rather than ammonium nitrate, dust, or background in origin), the bias between the OPC-N2 and MicroPEM disappeared (slope = 1.02), yet the error metric was among the highest in the four aerosol categories and was above the target value (Table S10). These findings suggest more research is warranted to explore how changing aerosol characteristics (both assumed and actual) impact optical particle sensor performance. Summary statistics for each performance assessment metric are given in Tables S8-S10 in Sect. 8 of the Supplementary Information.

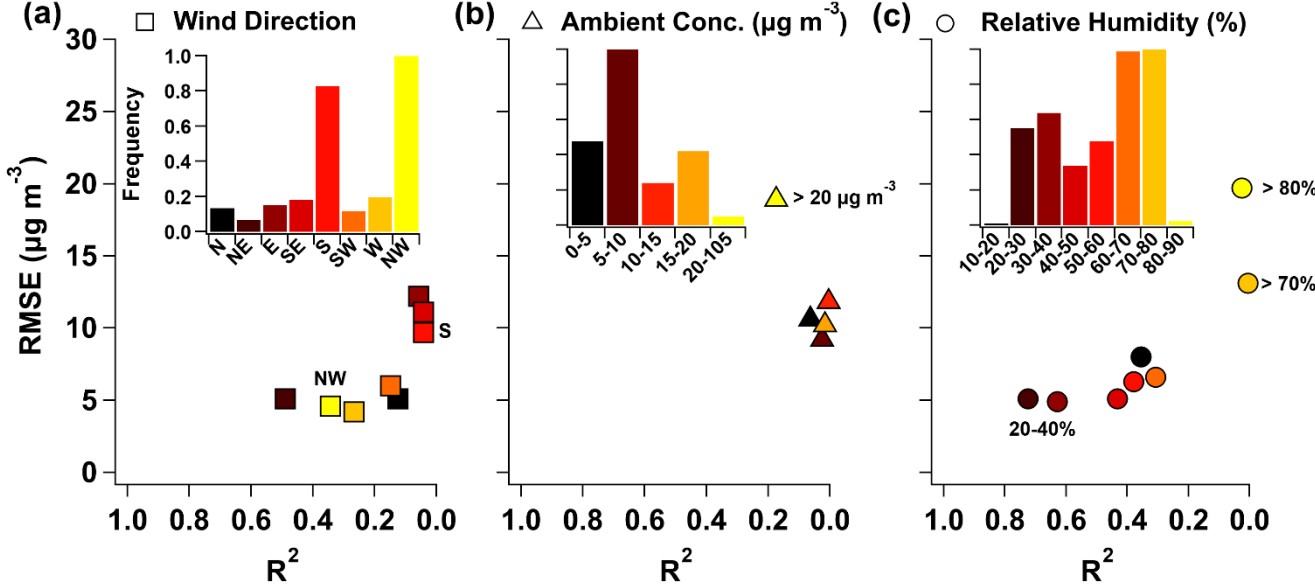

**Figure 4:** Performance comparison of the RH-corrected Alphasense OPC-N2 compared to the MicroPEM under different environmental conditions: (a) wind direction, (b) ambient concentration, and (c) relative humidity during collocation at the Village 2 site in Mulanje, Malawi. An individual data point represents the paired metrics (RMSE and $R^2$) for the OPC-N2 for a specific range of each condition. The histograms (inset) show the normalized frequency distributions for the ranges of each condition recorded during the collocation period. The colored markers in each panel correspond to the colored histogram bins. The metrics were calculated from 60-min averaged RH-corrected OPC-N2 PM$_{2.5}$ concentrations compared to the MicroPEM mass-corrected nephelometer. RMSE is root mean square error, assuming the MicroPEM concentrations as the true values; $R^2$ is the coefficient of determination. The lower left corner region of each panel indicates the highest performance based on these metrics.

In this deployment site, the OPC-N2 performed the best compared to the MicroPEM during dry conditions (20 to 40% RH) and when measuring background aerosol rather than source emissions (Fig. S14 - presumed based on time series data). However, this result might be partially due to the coincident effects of high RH (Fig. 7). Figure 4c shows OPC-N2 behaviour was affected by changes in ambient RH. In general, performance decreased with increasing RH, and this effect remained even after RH correction. For RH = 20 to 40%, RH-corrected OPC-N2 performance approached or exceeded the target values for the linearity, error, and precision metrics (Table S8). After RH increased past 70%, the $R^2$ value approached zero and the RMSE increased beyond the target value. Unfortunately, the inset histogram of Fig. 4c shows that an RH range of 60 to 80% was typical for this site during collocation.

We found that the OPC-N2 at this specific site underestimated mass concentration compared to the MicroPEM, based on less than unity slope values. The performance was variable at low ambient concentrations and largely dependent on RH (Fig. S13). However, outside of very humid (RH > 70%) or very dry (RH < 30%) conditions, the RH-corrected OPC-N2 could

estimate the PM$_{2.5}$ mass concentration within about 10 μg m$^{-3}$ of the MicroPEM value for real-time, hourly, and daily monitoring purposes (based on RMSE in Table S7). The findings from this section highlight the importance of quality assurance for low-cost optical particle sensor mass concentration measurements, especially those made in environments with highly variable meteorology and nearby ultrafine aerosol sources. For this site, contextual information on meteorology and emissions sources and their diurnal patterns helped interpret and evaluate the measurements.

### 3.3 Gas sensor performance during deployment

Given that RH, T, DP, and differential voltage were inputs to the calibration models, the ranges of these values during collocation in NC should mimic the ranges expected during deployment in Malawi. Otherwise, the model is required to extrapolate beyond its training bounds, which could lead to non-physical results (e.g., negative concentration values). Further, the performance assessment statistics derived from the collocation cannot be expected to hold for conditions far beyond those experienced during the performance characterization. Overall, the collocation and deployment settings exhibited a similar range of environmental conditions (Fig. S15-S16), but T and RH ranges in NC (15 to 40°C and 20 to 80%) were less extreme than in Malawi (10 to 45°C and 10 to 95%). While in Malawi, the ARISense experienced more time at lower temperatures (T < 25°C), lower gaseous concentrations (other than CO), and lower ambient pressure (5 to 15 kPa lower depending on the location). Although the ARISense were deployed at a higher elevation in Malawi than during the collocation in North Carolina (625 m vs 120 m above sea level), all models were built using the differential voltages (WE-AE) of each electrochemical gas sensor. Therefore, the pressure-related shifts in the WE and AE baseline were not expected to pose an issue to the calibrated Malawi data. The variation in pressure was within the operating range given on the sensor specification sheets (80 to 120 kPa) and was stated not to have long term impacts by the manufacturer (Alphasense FAQs, 2021). Further, others have shown no statistically significant change in electrochemical sensor sensitivity due to changes in pressure (Popoola et al., 2016). Even so, we did not have the laboratory chamber data to investigate this potential issue.

### 3.3.1 Bivariate histograms

Figure 5 shows bivariate distributions of T, RH, and gas sensor differential voltage data collected in NC and Malawi. In addition to capturing interactions between variables, Fig. 5 shows that even when in the same environment during the NC collocation, the individual sensors in each ARISense responded differently. Compared to ARI013 and ARI014, the O$_x$ sensor in ARI015 showed weaker temperature dependence (Fig. 5c). Since ARI015 had a shorter collocation period, it could be hypothesized that if ARI015 were present in the collocation environment for the same amount of time as ARI013 and ARI014, its response would look more like the ranges measured by the other sensors. However, this cannot fully explain the variation between individual sensors. For example, there is considerable variation between the ARI013 and ARI014 NO$_2$ differential voltage ranges (grey regions in Fig. 5g-h), despite having identical collocation periods. Further, the raw CO sensor data for all three monitors showed much less inter-sensor variation (grey regions in Fig. 5d-f), even despite the shorter collocation period of ARI015. This inter-sensor variation, which appears largest for the NO$_2$ sensors, may partially explain

the lower performance of this gas sensor group during calibration model performance testing, compared to the other gas sensor types (Fig. 2).

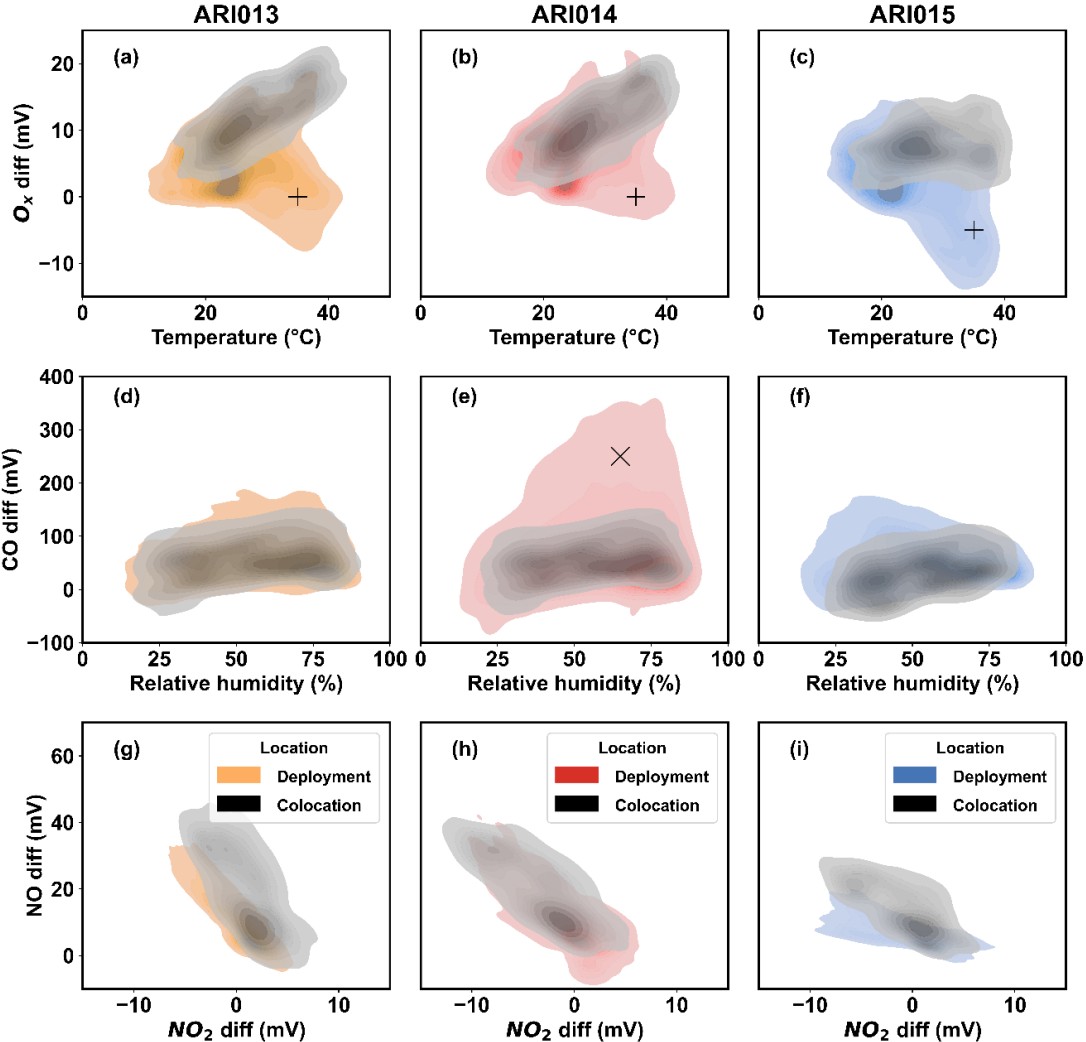

**Figure 5:** Bivariate distributions of gas sensor calibration model data inputs (RH, T, and $O_x$, CO, NO, and $NO_2$ differential voltage) for each ARISense monitor using kernel density estimation. Density is reflected in the color scheme; Darker colors indicate more data points in that region. Training data collected during collocation in North Carolina are shown in grey; data collected during deployment to Malawi are shown in color. ARI013 was deployed to the Village 2 site, ARI014 to the Village 1 site, and ARI015 to the University site. Regions where the deployment distributions overlap with the NC collocation distributions indicate the regimes for which the calibration models were trained. Regions where the deployment location distributions extend beyond the NC collocation distributions indicate regimes where the calibration models must extrapolate to estimate pollutant concentrations. These regions are indicated by overlaid markers 'x' and '+' and are discussed in the text.

There were notable regimes in Malawi that required the calibration models to extrapolate beyond NC training conditions. NO differential voltage responses in NC and Malawi did not completely overlap (Fig. 5g-i), especially in the low-concentration regime (i.e., V near 0 mV) which was more frequent in Malawi. The collocation site in NC was 10 m from an 8-lane freeway (Saha et al., 2018), therefore $NO_x$ concentrations were higher than in rural Malawi where vehicles and industry are rare. However, for ARI014 in Village 1, there was a higher $NO_2$ response in the deployment environment compared to the collocation environment. This could be partially explained by sensor interference by RH and T, which were more extreme (i.e., beyond the training ranges) in Malawi (Fig. S17). Figure 5e shows the maximum ARI014 CO differential voltage in Malawi (350 mV) was three times higher than the maximum voltage registered in NC (100 mV). This high CO regime is denoted by an 'x' on Fig 5e. This difference was consistent with observations of nearby sources (Fig. 1c-d). ARI014 was deployed in more densely populated Village 1, adjacent to more biomass cookstove activity than ARI013 or ARI015 (Fig. 1c). In general, we expected higher CO in Malawi than in NC, where biomass burning is less common and emissions from other sources (e.g., vehicles) are controlled by strict federal regulation.

The $O_x$ differential voltage ranges were the most dissimilar between the collocation and deployment environments. The most frequent regimes, the heaviest shaded regions in Fig. 5a-c, did not fully overlap for any of the ARISense. In NC, the relationship between the $O_x$ sensor voltage and ambient temperature was positive and monotonic. Higher temperatures generally facilitate ozone production, therefore this relationship fit our expectation for an urban site in a single season. However, the positive relationship between $O_x$ sensor voltage and temperature did not always hold in the deployment sites. Figure 5a-c shows a high temperature-low ozone regime in Malawi (regions denoted by a '+' marker) that was not present in the NC data. Further, for all three Malawi sites, the minimum $O_x$ sensor voltages were lower ($-10 < V_{min} < 0$) than minima in the NC collocation.

### 3.3.2 Diurnal trends

Since the deployment sites did not have reference data for quantitative comparison, we calculated and compared the annual mean diurnal trends of each pollutant at each site, as predicted by the five models to qualitatively assess the transferability of the calibration models to Malawi. Our definition of a transferable model required that it produce: (a) non-negative concentration values and (b) diurnal trends consistent with our first-hand observations of nearby emission sources and their timing, previous observations of diurnal trends in regions with widespread biomass cookstove use (Dionisio et al., 2010; McFarlane et al., 2021; Subramanian et al., 2020), and knowledge of atmospheric chemistry. Non-physical predictions from a given model may indicate that differences between the collocation and deployment environments were too large to extrapolate and therefore any deployment results calibrated by that model are likely not reliable. Alternatively, coherency among the concentration values and trends estimated by the models may suggest that the deployment results are robust against variation in the modelling approaches. This analysis can contribute to our confidence in the estimated concentration

values and trends, but cannot address or estimate the quantitative error. Diurnal trends in Figure 6 suggest the kNN hybrid model was the most transferable for interpreting deployment data for all gas sensors. However, both the kNN and RF hybrid models predicted similar trends and values for most sensors. The MLR and HDMR models also predicted similar trends, but frequently predicted negative values.

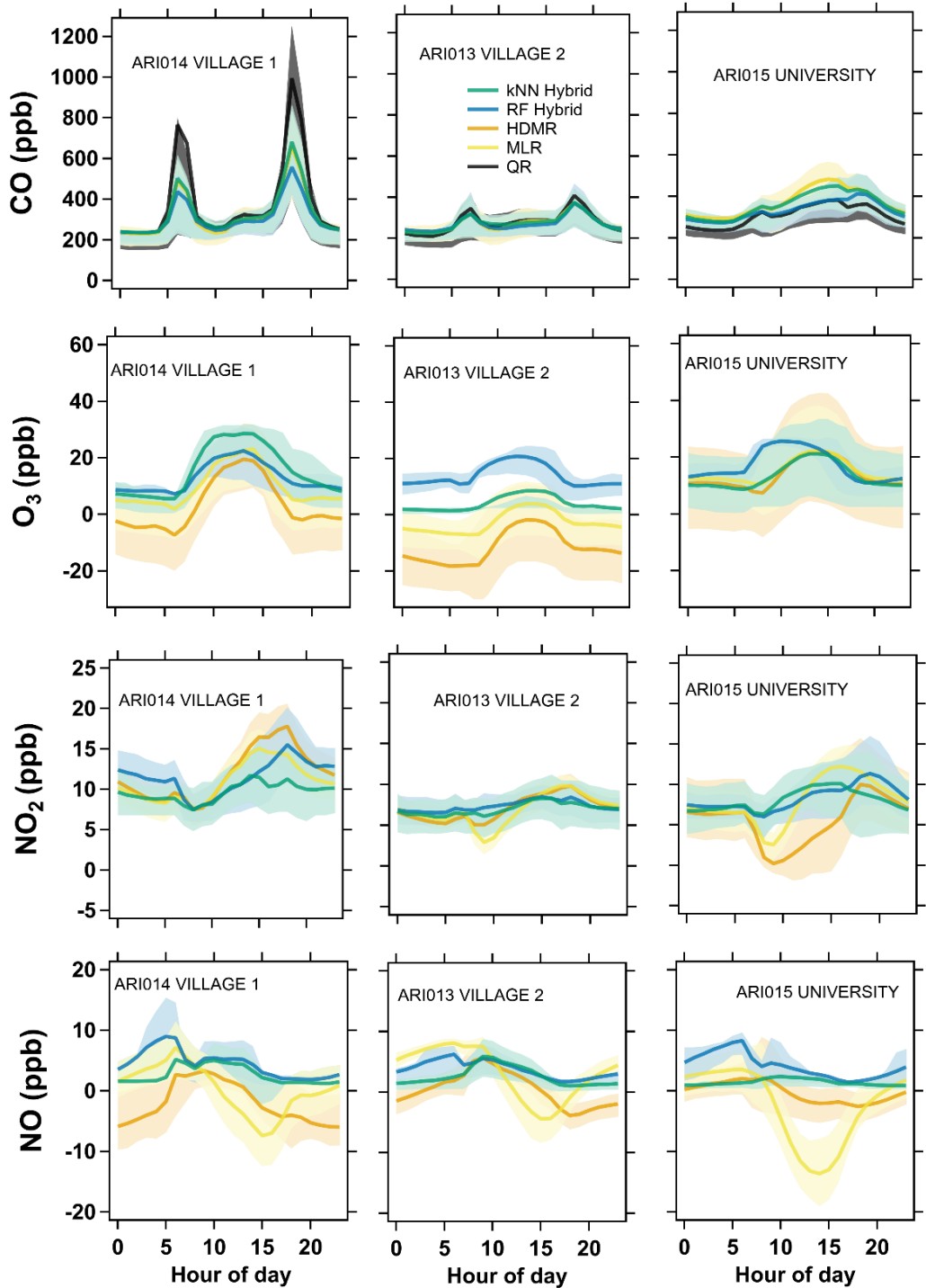

**Figure 6:** Diurnal trends of calibrated gas measurements (rows) at each site (columns) in the three deployment environments. QR model built for and applied to CO data only. The thick line indicates hourly mean, the shaded region indicates interquartile range. Midnight is the zero hour. The hours are in local time.

Calibrated CO data showed the highest coherency across model predictions and were rarely non-physical (Fig. 6). All models predicted similar diurnal trends, specific to each site. Knowledge of the nearby emission sources and activity patterns lend support to the calibrated CO data. For example, the village monitors were adjacent to widespread household biomass cookstove activity, coincident with the concentration peaks seen in the diurnal data. This diurnal cooking pattern was observed in both CO and OPC-N2 data (Fig. 6 and Fig. 7, respectively) at both village sites and was measured in complementary emissions monitoring work (Bittner et al, in prep). Further, ARI014 was in a more densely populated village than ARI013, contributing to higher CO peaks (Fig. 1c). The QR model overestimated CO peaks compared to other models for the Village 1 data, likely because the model training set did not include high concentration data (Fig. 5e) and the quadratic term was not well constrained. Despite the calibrated CO measurements in Malawi being higher than the concentrations experienced in NC, particularly for ARI014 in Village 1, we expect that the calibrated CO measurements from Malawi are credible (excluding the QR model). We provide the following reasons for justification: a) the manufacturers report that the sensor response is expected to be linear up to 500 ppm (Alphasense, LTD., 2019), b) RH/T interference induced on the CO-B4 sensor, approximately 0.2 mV/ppb (Lewis et al., 2016), has relatively less influence on overall sensor readings in the higher voltage (i.e., concentration) regime c) all modelling approaches (other than QR) predicted highly similar diurnal trends and concentration values, and d) there were known CO emission sources, with diurnal usage patterns matching the observed trends, near the monitoring sites. This suggests, for this specific sensor under these conditions, that these modelling approaches (other than QR) could reliably extrapolate beyond the training data limits to provide reasonable measurements in the deployment environment.

The calibrated $NO_x$ data showed less coherency than the CO data. $NO_2$ trends were similar across the sites and concentrations were rarely negative, but calibrated NO trends varied across models and the lower performing models (HDMR and MLR) often predicted negative values. The better models identified in the NC collocation, kNN and RF hybrid, suggested that mean ambient $NO_x$ levels in Malawi were low (< 15 ppb). We have lower confidence in the calibrated $NO_x$ measurements in Malawi for the following reasons: a) the calibrated observations (5 to 20 ppb) were on the same order of the noise level reported on the sensor specification sheets (15 ppb) and b) the lack of coherency observed between model predictions. Low ambient $NO_x$ levels and a lack of representative data in the NC collocation data likely contributed to the non-physical concentrations predicted by the models in Malawi.

The calibrated $O_x$ sensors performed the best during collocation testing compared to the other gas sensors, but in Malawi the calibration models frequently returned non-physical values and showed inconsistent annual diurnal trends between the models and across the sites. For ARI014 and ARI015, the $O_3$ trends were consistent in shape and magnitude and were aligned with the expected diurnal trend (i.e., peaking at midday). Peaks in the mean concentration were between 10 and 30 ppb, plateauing from 10 AM and 3 PM local time. The RF hybrid model at the ARI015 University site estimated the $O_3$ peak to occur earlier in the day compared to the other models and sites. This may be the result of a spurious relationship between

$O_x$ voltage and DP in the collocation data set on which the RF Hybrid model was trained, which held at the Village sites but not at the University site. At the Village 2 site (ARI013), there was a change in raw differential voltage response after December 2017 that caused all models to fail for the second half of the deployment. All models either consistently predicted
negative values, values < 1 ppb, or failed to reproduce the expected diurnal trend (i.e., peaking around 9am rather than midday). Only $O_x$ data collected before December 2017 resulted in reasonable calibrated values and trends (Fig. S18). Notably, $O_x$ data collected after December 2017 corresponded with the high temperature-low ozone regime (Fig. S19) shown in Figure 5a-c. Despite the $O_x$ differential voltage data spanning a similar range in both NC and Malawi, there was little overlap in the ozone dimension at comparable concentration, RH, and T conditions. Since ozone is a secondary pollutant
driven by complex atmospheric processes and multiple precursors, the ambient conditions that increase or decrease ozone formation in one region may not hold in another environment. Although the calibrated $O_x$ sensors performed better than the other gas sensors in NC, the models were tuned for a set of conditions that did not hold in Malawi. This suggests that for these $O_x$ sensors and these modelling approaches, a lack of environmentally similar collocation data compromised our ability to reliably interpret calibrated $O_3$ measurements in this specific deployment environment.

**3.4 OPC-N2 performance during deployment**

To evaluate the long-term performance of the OPC-N2 during deployment in Malawi, we examined the representativeness of the collocation conditions for the full year of conditions experienced during deployment. Figures S20-S21 show normalized histograms of the T, RH, and $PM_{2.5}$ mass concentration observed during the collocation and the full-year deployment in Malawi, suggesting the two data sets spanned a similar range of environmental conditions. However, the collocation
occurred during the cool, dry season, and RH minima and maxima (regimes associated with deficient performance during collocation – see Section 3.2) were more extreme during the 1 year deployment in Malawi.

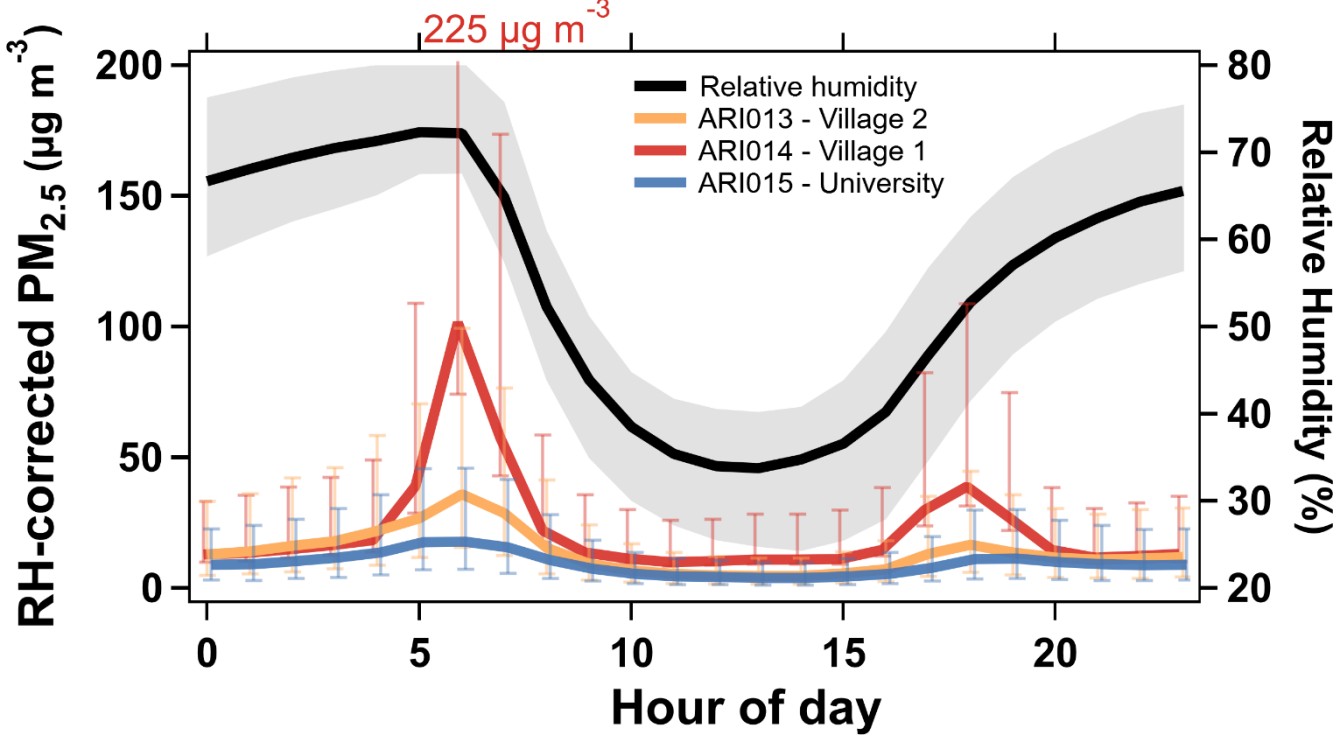

**Figure 7:** Diurnal trends of the integrated mean PM$_{2.5}$ mass concentration measured by the OPC-N2 in each ARISense at each deployment site (left axis) and the annual relative humidity at the Village 2 site (right axis). Error bars represent the calculated 1σ (68%) prediction interval of the hourly mean value. The red text annotation indicates the upper limit of the Village 1 prediction interval at 6 AM (beyond the range of shown y-axis). Thick lines indicate hourly mean and shaded regions indicate interquartile range.

Figure 7 shows the annual diurnal trend of the mean PM$_{2.5}$ mass concentration, with 1-sigma prediction intervals, using hourly-averaged, RH-corrected data from each deployment location. Peak PM$_{2.5}$ concentrations were observed around 6 AM local time at all sites, when morning biomass cookstove activity coincided with high RH (and more atmospherically stable) conditions. Figure 6 shows that the diurnal trends of ambient CO (another pollutant emitted by biomass burning) were similar to the PM$_{2.5}$ diurnal trends at each site. Again, the largest peaks were observed at the more densely populated ARI014 Village 1 site. The prediction intervals were widest between 5 and 7 AM local time, indicating overall low confidence in OPC-N2 measurements during this period. Afternoon and overnight means, coinciding with drier conditions, were similar across all three sites and prediction intervals were narrowest during afternoons. Data from the more remote locations (ARI013 and ARI15) suggest background concentrations of PM$_{2.5}$ in rural Malawi were low (5 to 15 µg m$^{-3}$), but the OPC-

N2 could not reliably quantify peak concentrations that were high and variable, dependent on the nearby sources and covariance with ambient meteorology (RH). Despite this, qualitative data from the OPC-N2 sensors was sufficient to identify nearby source activity and indicate periods when ambient concentrations were likely high enough to be harmful to human health (and at least partially driven by cooking activities associated with higher exposure concentrations).

**3.5 Comparison of ARISense CO to remote sensing and reanalysis data**

Given the absence of additional in-situ surface data, we rely on satellites and models to estimate surface air quality for comparison of our results. To contribute to the literature on surface-to-satellite comparisons over Africa, we compared calibrated ARISense CO observations to a satellite observation (MOPITT) and a model estimate (MERRA-2) in our study region. We confirmed that all three data sets reported similar annual qualitative trends, although they disagreed in magnitude. This analysis was limited to CO, given that the calibrated CO observations were the most dependable of the

ARISense gas data and NASA remote sensing data products were more readily available for CO compared to $O_3$ or $NO_x$.

Figure 8 shows the mean monthly CO from the University (ARI015) and Village Mean (average of ARI013 and ARI014) sites compared to that from two area-averaged remote sensing products: CO surface mixing ratio from MOPITT and CO surface concentration from MERRA-2. All three data sets were compared from July 2017 to July 2018, focusing on

differences between the peak agricultural burning (Sept to Oct) and non-burning (Dec to Jul) seasons. November and August were excluded from either description (peak burning or non-burning) for the following reasons: (a) a review of fire studies in the region consistently reported Sept and Oct as the dominant months of the burning season (Nieman et al., 2021), (b) Aug and Nov mark the beginning and end of the fire season, respectively, therefore cannot be considered non-burning months, (c) the exclusion of Aug and Nov better captures strong seasonal differences, providing a measurable benchmark to compare the

satellite and surface data, and (d) ARISense data for the Village sites was unavailable for Nov 2017 (see Sect. 3.7 - on difficulties in deployment).The MERRA-2 data set was complete for the full year of interest, but MOPITT was missing data for the Village Mean region in February and March 2018. The remote sensing data sets were more similar to one another at the Village Mean site compared to the University site. At both sites, MOPITT reported higher CO concentrations than MERRA-2, especially in the peak burning season.

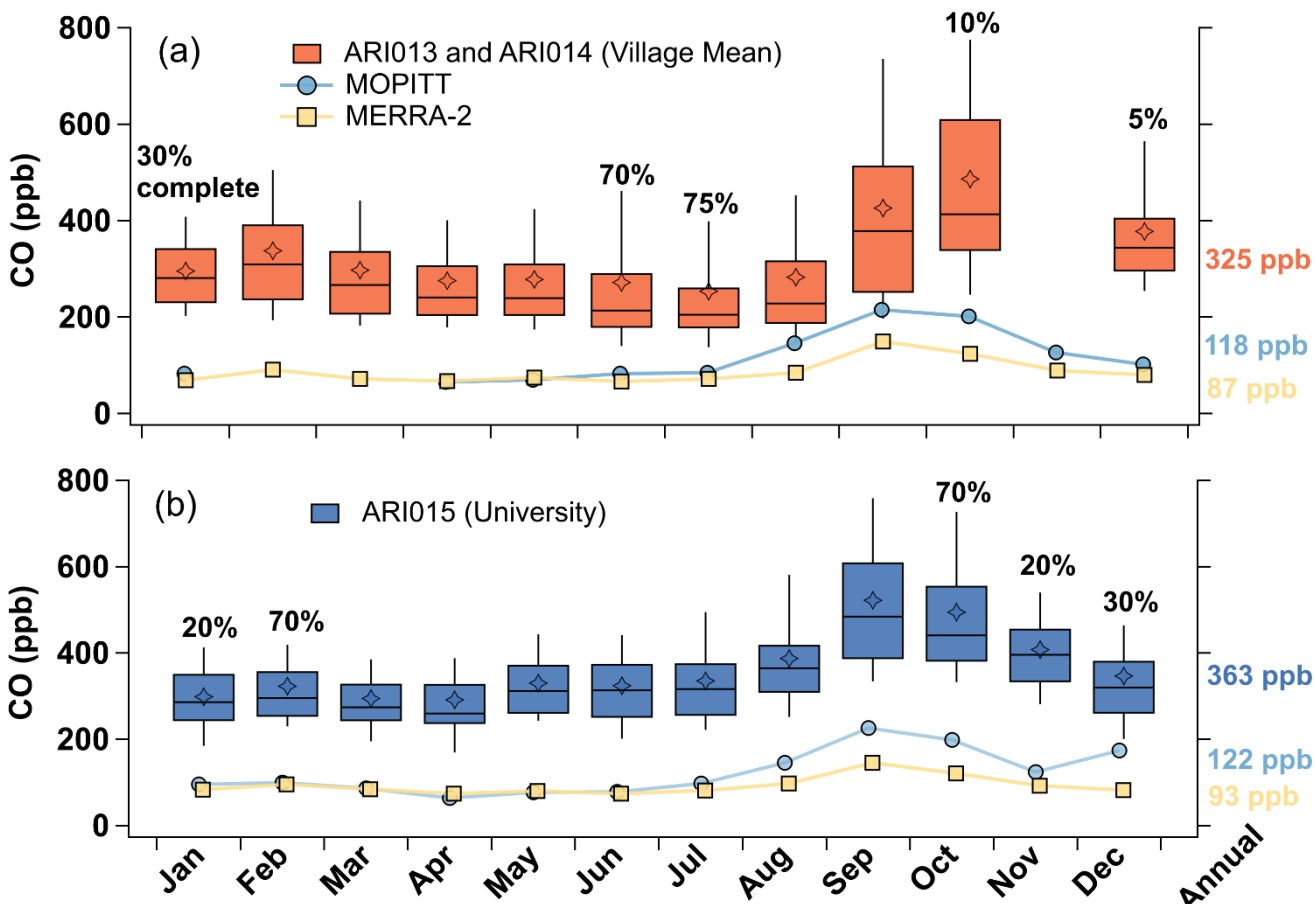

**Figure 8:** Monthly carbon monoxide (CO) concentration (ppb) reported by the surface ARISense (Tukey box plots) and remote sensing data products (lines and markers indicating mean monthly value) at the (a) Village Mean and (b) University sites. Top and bottom of boxes indicate 75th and 25th percentiles, whiskers show 9th and 91st percentiles, midline indicates median, and stars indicate mean. The ARISense surface data were at least 80% complete for each month except where noted with a percentage text label. Data for July 2017 and July 2018 were averaged. Village Mean represents the average of ARI014 (Village 1) and ARI013 (Village 2) data. The annual mean from each data source is given on the right axis. MOPITT (Multispectral CO Surface Mixing Ratio Daytime/Descending) is a satellite measurement; MERRA-2 (CO Surface Concentration -ENSEMBLE) is a global reanalysis product.

All three datasets (MOPITT, MERRA-2, and ARISense) indicated that annual mean CO concentrations were slightly higher overall at the University site than at the Village site, although this was less pronounced in MERRA-2. Similarly, all three data sets showed increased ambient concentrations during the peak burning season compared to the non-burning season at both sites. For ARISense, MOPITT, and MERRA-2 observations, respectively, peak season means were larger than non-

burning season means by 160 ppb, 130 ppb, 60 ppb (Village Mean) and 190 ppb, 115 ppb, 50 ppb (University). Although the ARISense indicated larger absolute differences between seasons, the relative increase at both sites was only about 50% of the non-burning season mean, while MOPITT and MERRA-2 reported increases of 125% and 75%, respectively. This could be explained by ARISense proximity to small-scale combustion activity not resolved by satellite imaging. Satellite-based observations approximate ambient background concentrations, which increased during the peak season due to regional agricultural burning. Meanwhile, the ARISense were exposed to ambient background concentrations as well as nearby biomass cookstove emissions, which presumably remained consistent throughout the year, showing a lower relative seasonal increase during the peak burning season. Quantitative disagreement between surface and remote CO observations was highest during the burning season, especially at the University site (Fig. 8). Remote sensing data suggested higher CO concentrations at the University compared to the Village Mean during non-burning periods, but during the peak burning season this difference shrank and similar concentrations were observed across both sites. Conversely, differences between ARISense observations grew by about 6% during the peak season. MERRA-2 and MOPITT concentrations were highest in September, consistent with ARISense data at the University site, but not the Village Mean site which peaked in October. However, 90% of the October CO data were missing for the Village site.

Monthly mean CO ARISense values were 2 to 4 times higher than those reported by MOPITT and MERRA-2. We found differences of 175 to 200% between the annual mean CO concentration from ARISense and MOPITT, depending on the site, and even larger differences (up to 360%) with MERRA-2. Differences between MOPITT and MERRA-2 were smaller (30 to 35%). There are few comparable studies available to explain these differences, which are greater than previously reported in the literature available for SSA. One study in South Africa reported relative differences of ±40% between ground-based CO measurements and Aura satellite observations at Cape Point station (Toihir et al., 2015). Many studies found good agreement (within 10-20% bias) between ground measurements and MOPITT observations, but this was for Total Column CO, and the observations were not limited to comparisons over Africa (Buchholz et al., 2017; Emmons et al., 2009, 2004; Yurganov et al., 2008, 2010). However, these studies found negative satellite bias when intense biomass plumes affected observations, when CO levels were low in the Southern Hemisphere, or when atmospheric CO levels changed rapidly (Buchholz et al., 2017; Emmons et al., 2004; Yurganov et al., 2008, 2010). Each of these conditions could be expected to occur in the southern Africa troposphere, potentially explaining differences observed between the ARISense and remote sensing observations in this study.

This comparison of low-cost sensor surface data, satellite observations, and model estimates in Malawi suggests each of these resources can give consistent information on qualitative, long-term trends in a region without ground-based reference monitoring. However, because of inherent differences in spatial and temporal resolution, each observation will disagree in magnitude. Satellite retrievals and real-time surface measurements do not result in directly comparable quantities. Satellite data are collected as a once-daily flyover observation, averaged over a ~12,000 square kilometer area (corresponding to 1°

spatial resolution). In contrast, the ARISense data were 1 min resolution, fixed-site, long-term point measurements at the surface. Further, the ARISense data were collected near visually identified biomass emission sources and were not representative of background conditions. Meanwhile, the satellite observations provide an estimate of regional background conditions. Despite these differences, the MOPITT, MERRA-2 and ARISense data sets agreed on the long-term seasonal trends present in this region, and even corroborated site-to-site differences (e.g., higher mean CO at University compared to Village Mean site). These findings suggest the ARISense captured synoptic-scale variation in CO, but comparison to remote sensing data does not allow for a quantitative assessment of data collected at higher temporal resolutions.

**3.6 Comparison to other ambient measurements in SSA**

The annual median (July 2017 to July 2018) ARISense surface concentrations estimated by the ARISense sensors were 9 to 11 ppb for $NO_x$, 4 to 15 ppb for $O_3$ and 240 to 330 ppb for CO, depending on the site. Surface concentrations and diurnal trends of ARISense CO and PM in Malawi were comparable to studies in Kenya, Rwanda, Ethiopia, Uganda, and South Africa (Delmas et al., 1999; DeWitt et al., 2019; Laakso et al., 2008; McFarlane et al., 2021; Nthusi, 2017; Scheel et al., 1998; Subramanian et al., 2020; Toihir et al., 2015). However, comparison of $O_3$ concentrations suggested the calibrated ARISense observations underestimated actual concentrations. ARISense $NO_x$ observations were similar to two other studies (Delmas et al., 1999; Laakso et al., 2008), but overall, there is little comparable data available to assess $NO_x$ concentrations in Africa.

ARISense CO observations were similar to regional CO concentrations in Central Africa (measured by aircraft), found to be in the range of 250-400 ppb (Delmas et al., 1999). A long-term ambient study at the Rwanda Climate Observatory found a mean CO concentration of 215 ppb from May 2015 to January 2017 (DeWitt et al., 2019), only slightly lower than our findings in Malawi. Another LCS study in Kigali, Rwanda observed a range in ambient CO concentrations, from 225 to 500 ppb at their rural and urban sites (Subramanian et al., 2020), spanning the concentration range we observed at our rural and semi-urban sites in Malawi.

Both studies of Rwanda found mean ambient $O_3$ concentrations of 30 to 40 ppb (DeWitt et al., 2019; Subramanian et al., 2020). For a "relatively clean background site located in dry savannah in South Africa the annual median (July 2006 to July 2007) trace gas concentrations were equal to 1.4 ppb for $NO_x$, 36 ppb for $O_3$ and 105 ppb for CO" (Laakso et al., 2008). Background levels of $NO_x$ and CO at this site were 2 to 5 times lower than the ARISense annual means, yet background $O_3$ was in line with the Rwanda studies. This suggests regional ozone concentrations in Central and Southern Africa are presently about 30-40 ppb. The annual mean ARISense $O_3$ values were up to a factor of ten lower, however, we identified quality assurance issues in the calibrated $O_3$ values, particularly for the second half of the deployment data, therefore the ARISense data are likely to be an underestimate of the true ambient values.

This South African 'clean' background site had $NO_x$ concentrations up to a factor of 10 lower (1.4 ppb) than ARISense measurements in Malawi (Laakso et al., 2008), but aerial measurements made during intense savanna fire activity in Central Africa found $NO_y$ present in the range of 4-10 ppb (Delmas et al., 1999). Together, these studies suggest that the ARISense NOx concentrations (9-11 ppb) may be reasonable for our non-background, biomass emission influenced sites in Malawi.

Notably, the corresponding $PM_1$, $PM_{2.5}$ and $PM_{10}$ median concentrations at the clean South Africa background site: 9.0, 10.5 and 18.8 µg m$^{-3}$, respectively (Laakso et al., 2008), were comparable to ARISense observations. The annual median ARISense RH-corrected $PM_1$, $PM_{2.5}$ and $PM_{10}$ concentrations were 4 to 7, 6 to 10, and 13 to 20 µg m$^{-3}$, respectively, depending on the site. It is possible that actual concentrations of fine PM were higher at the sites in Malawi, given that concentrations of gaseous emission tracer species (i.e., CO, $NO_x$) were higher compared to regional background levels found by other studies. However, given the high minimum cut-off diameter of the OPC-N2, this particle sensor would have been unable to detect ultrafine particles emitted from biomass burning. Average ambient $PM_{2.5}$ concentrations (measured with an Alphasense OPC-N2) were found to be 11 to 24 µg m$^{-3}$ at various sites in Kenya, with higher pollution episode concentrations ranging from 35 to 51 µg m$^{-3}$ (Nthusi, 2017). Median ARISense $PM_{2.5}$ concentrations were also comparable to U.S. embassy measurements in Ethiopia and Uganda (DeWitt et al., 2019). Taken together, these comparisons suggest PM levels in rural Malawi are comparable to regional measurements made across SSA, but localized impacts from biomass cookstoves can result in higher concentrations of fine PM, which are difficult to accurately quantify with the OPC-N2. In all, although these comparisons are not a substitute for quantitative evaluation of the ARISense in Malawi, they provide a benchmark for comparison and suggest that the CO, $NO_x$, and PM ARISense observations are reasonable for this region. At the same time, they cement our conclusion that ARISense $O_3$ observations are likely erroneous for this environment.

## 3.7 Performance of ARISense sensor packages over time

Total data recovery for the 1 year deployment varied by site, season, and sensor, with rates ranging from 30% to 80% (Fig. S22). Average recovery for the 1 year deployment was around 60%, with highest recovery at the University site (80%) and lowest at Village 1 site (40%). Data across all sites had the highest completeness (>70%) in the cool-dry (Jun-July-Aug 2017 and 2018) and the cool-wet season (Mar-Apr-May 2018). Data losses were mostly explained by power outages, software failures, and sensor equilibration times required after a power outage (Fig. S23). Power outages were common in the warm-wet season (Dec-Jan-Feb) due to insufficient solar intensity resulting from extended periods of heavy cloud cover. At the ARI014 site, insufficient power led to an unanticipated diurnal cycle wherein the monitor would shut off in the early morning hours and require a few hours of solar power before turning on again. This daily cycle, coupled with the 8-hour long NO sensor re-equilibration time, led to almost 0% NO data recovery in the second half of the deployment for Village 1. In all, nearly 50% of data losses at the ARI014 site were due to insufficient power or failure to write data to file. Corrupt USB storage devices, which we were slow to replace due to ongoing civil unrest (The Guardian, 2017), resulted in significant data losses in the hot, dry season (Sept-Oct-Nov) at the two Village sites. Individual sensor failure was rare, but

two months of ARI014 $O_x$ data were lost to electrochemical sensor drift and one OPC-N2 (ARI013) failed in the last 3 months of deployment due to an insect nest clogging the OPC-N2 inlet. In all, we recorded 6992 hours of data at the University site (ARI015), 5860 hours for Village 2 (ARI013), and 4720 hours for Village 1 (ARI014). Future deployments should include insect screens over all sensor inlets and improved battery storage and power systems that run at a longer duty cycle in the case of insufficient solar (e.g., power on only once battery is fully charged) to minimize the impact of sensor equilibration times on data recovery.

Since the monitors were deployed to their sites for >1 year, there was observation overlap in seasonally similar data collected one year apart. To gain insight into sensor stability, we compared the data collected in the first month (July 2017) to the final month (July 2018) of the deployment, given that ambient environmental conditions were similar in July of both years (additional details in Sect. 11 of the Supplementary Information). It is not possible to know if the range of gas concentrations were significantly different between July 2017 and July 2018. We explored this analysis on the assumption that inter-annual variability in ambient concentrations was minimal. Bivariate distributions of the raw differential voltage readings from July 2017 and July 2018 showed that the most frequent observations (i.e., heaviest shaded regions) were approximately the same in both years (Fig. S25). Observable differences in the voltage measurements could be partially explained by known environmental differences. For example, the $O_x$ sensor voltages in July 2018 were lower on average than in 2017, but this was consistent with lower temperatures and higher RH in 2018 compared to 2017. However, there was potential evidence of slightly reduced or altered responses in individual sensors, particularly the NO sensors in ARI013 and ARI015 and the CO sensors in ARI013 and ARI014. For these sensors, the 2018 distributions had less spread than the 2017 distributions, suggesting either less variation in ambient concentrations in 2018 or decreased sensitivity in the sensors. Diurnal plots from both years showed that the raw mean voltages and trends were consistent (Fig. S26). However, again the most noticeable differences were in the individual CO and NO sensors identified from the bivariate distributions. For example, the CO peaks measured at mealtimes by ARI013 and ARI014 were about 50 mV lower in 2018 than 2017. These differences could be explained by lower concentrations in 2018 than 2017, changes in the raw sensor response over the one year period, or by both. Without reference equipment, we were unable to investigate sensor drift and decay more rigorously. This qualitative analysis suggests individual sensor responses were altered during the one year deployment, but there was no unambiguous evidence for systematic deterioration within or across the electrochemical sensor groups used in the ARISense.

In general, the calibrated observations followed the trends identified from the raw sensor voltage readings. Calibrated CO data trends were consistent for both years, with the models responding as expected to the lower voltage readings in 2018 compared to 2017. For ARI013 and ARI014, the calibrated CO peaks at mealtimes were accordingly lower, by about 100 ppb, in 2018 (Fig. S27). However, although the raw $O_x$ sensor trends in 2018 and 2017 were consistent for all the ARISense (Fig. S26), the kNN hybrid model calibrated $O_3$ data were highly irregular between the two years (Fig. S27). For example, the calibrated $O_3$ data for July 2017 showed the expected diurnal pattern (concentration increasing with solar intensity) with

plateaus between 15 and 40 ppb, depending on the site. Yet in July 2018, although the raw $O_x$ diurnal data looked similar to 2017, the calibrated data for ARI013 and ARI015 showed noon-time values between 0 and 5 ppb, and the diurnal trend for ARI013 showed a flat line (i.e., not correlated with solar activity). This finding, that raw $O_x$ sensor voltages were similar year to year while the calibrated $O_3$ values were not, provides further evidence that the lack of comparable T/RH/ozone collocation data contributed to the non-physical $O_3$ trends observed during the second half of the deployment at the ARI013 and ARI015 sites.

Before their return to NC, ARI013 and ARI014 were used for high-concentration emissions monitoring experiments after the one year ambient monitoring campaign was completed (Table 2). The reference monitor data from the post-deployment collocation in NC (Aug 2018 to May 2019) were intended to enable investigation of changes in ARI013 and ARI014 raw sensor response and model performance. However, the resulting data instead demonstrated that sensors had been severely degraded during the high-concentration exposures. In the post-collocation data, the raw differential voltage gas sensor responses in ARI013 and ARI014 were well correlated with each other ($R^2$ = 0.7 to 0.9) (excluding the ARI013 $O_x$ sensor which was clearly degraded: Fig. S28), but less correlated than during the pre-collocation comparison ($R^2$ = 0.9 to 0.99). To facilitate comparison with the pre-collocation performance metrics shown in Fig. 2 and Tables S4-S6, the performance metrics for the post-deployment collocation are given in Table S11 and S12. Despite showing inter-sensor consistency, the raw differential voltage sensor measurements (other than CO) made by ARI013 and ARI014 were poorly correlated with reference measurements (Fig. S29-S30). Inspection of the time series showed that the ARISense NO sensors tracked some spikes in the time-aligned NO reference data, but the $NO_2$ and $O_x$ sensors did not track reference data trends (Fig. S31-S32). The time series of the differential voltage and temperature data suggest the gas sensors in ARI013 and ARI014 were responding similarly to changes in T and RH, but they were no longer sensitive to changes in the target gas (Fig. S31). This may explain why the sensors in ARI013 and ARI014 were still well correlated with each other, but why they were not correlated with reference measurements. The calibrated CO data were the only data still roughly correlated with CO reference measurements, although the calibrated CO data showed aberrant features (Fig. S33-S34). These ambient sensors (except for the CO sensor) were likely affected by high concentrations of PM and volatile gases (e.g., hydrocarbons, formaldehyde, etc.) co-emitted during the biomass burning experiments. Exceedingly high concentrations of emissions can chemically degrade or contaminate the sensors, for example, the catalyst or electrolyte can be affected or depleted by repeated interactions with high concentrations of non-target species emissions. Further, if there were high concentrations of fine volatile PM permeating the inlet and flow line, it could condense and block or attenuate the sample flow rate. The $O_x$, NO, $NO_2$ sensors were permanently altered by the biomass burning emission experiments in Malawi, leading to poor performance during post-deployment collocations with reference instruments in NC. Given these dramatic changes in sensor responses, the models were unable to generate reasonable concentration values from sensor signals and consequently, we were unable to use the post-deployment collocation data set to quantitatively assess long-term model performance. The

partial exception to this was for the kNN hybrid calibrated CO data, which was correlated with the reference data ($R^2 = 0.5$), suggesting that the CO sensors might retain some function after additional collocation and recalibration.

## 4 Conclusions

Our experience showed that LCS networks are a viable method to collect novel surface AQ data in regions without reference equipment, but this approach requires strict data quality procedures to ensure the conclusions drawn from the resulting data are valid. Performance assessment in NC suggested the calibrated ARISense sensor packages (excluding the $NO_2$ sensor) would be suitable for supplemental air monitoring, based on U.S. EPA metrics and target values. However, performance during the pre-deployment NC assessment did not reflect performance in Malawi. For this deployment site, we found that detailed information about nearby sources and their diurnal emission patterns, ambient meteorological data, and a familiarity with air pollutant behavior were helpful when qualitatively assessing LCS performance in a region where quantitative assessment was not an option. A lack of coherency in diurnal trends between calibration model predictions and frequent non-physical concentration values (Fig. 6) showed that LCS measurements made in deployment environments different from the collocation environment can be unreliable and may lead to biased information about the deployment environment. For example, although the $O_x$ sensors showed the highest performance of all sensor types during collocation testing, and the measured RH, temperature, and $O_x$ voltage ranges were similar in the collocation and deployment environments, the calibrated $O_3$ data in Malawi were unreliable. The collocation data were collected in an urban area near a highway and the deployment data were collected in a rural area heavily impacted by biomass burning emissions. This difference in ozone precursor emissions could have contributed to the deficient performance of the calibration models in the deployment environment. We expect our experience in Malawi may generalize to other regions, suggesting that additional research is needed to address the issue of LCS calibration for secondary pollutants.

We found that the kNN hybrid modelling approach performed the best in the NC and when applied to data collected in Malawi. However, the general lack of standardization in LCS calibration and assessment approaches complicated and extended the calibration process for our study. Although there have been advancements in calibration methods, the difficulty of identifying and applying a singular best calibration model remains a common issue among LCS users (Topalović et al., 2019; Lewis and Edwards, 2016; Giordano et al., 2021). From an end user perspective, the burden of calibration easily becomes overwhelming. There is presently no clear guidance on which model would be appropriate for which sensor under which circumstances. This limits the potential user base of LCS technologies, complicates our ability to generalize findings across different studies, and may even to inferior quality measurements. Given the wide range in potential LCS technologies and deployment conditions, it is not possible to fully generalize the viability and sensitivity of the ARISense to another LCS package deployed in a different area. Nonetheless, we surmise LCS are most useful when they are carefully selected and calibrated for a single purpose and location, for which the environmental and pollutant conditions are at least partially characterized.

This pilot deployment also provided lessons regarding the design and deployment of low-cost AQ monitoring systems for off-grid applications. The ARISense packages survived the 1 year deployment to Malawi and enabled collection of a large, novel dataset, however they suffered individual sensor failures and frequent power losses. Given that 20 to 50% of the deployment data were lost due to insufficient power and corrupt data storage systems, for future solar-powered deployment efforts we suggest that the power system be designed to allow for primary and secondary data recovery goals (i.e., a back-up plan to prioritize the most desirable data in the event of insufficient power). Further, we were frequently restricted in troubleshooting and repair operations by spotty cellular connection, limited human resources, and our inability to remotely locate and procure appropriate equipment. A repair kit with basic equipment (e.g., pre-programmed USB devices, alternate SIM cards, hand tools with attachments specific to each LCS) stored in a nearby, secure location would have allowed for quicker troubleshooting and repair. We suggest that in addition to solar power limitations, other potential confounding factors like extreme weather and limited technical capacity and assistance availability be considered before deployment to remote locations. We found that the more closely located the monitor was to a trained local assistant, the lower the overall data losses were.

The responses of the LCS were not remarkably different after one year of deployment (Fig. S26-S27), assuming actual concentrations did not vary significantly from 2017 to 2018. However, except for CO, repeated exposure to high-concentration biomass emissions completely degraded the sensors. Key manufacturer specifications indicated that the CO sensor was the most robust. The CO sensor exposure limit was forty times higher than that of the $O_x$, NO, and $NO_2$ sensors. Further, the maximum temperature and RH range for the CO sensor was 50°C and 90%, respectively, and only 40°C and 85% for the $O_x$, NO, and $NO_2$ sensors. During deployment, the maximum ranges were occasionally exceeded for every sensor except CO. Operation beyond specified conditions, combined with ~100 hours of exposure to high concentration gases during the post-deployment emissions monitoring experiments, damaged the three less robust sensors (NO, $NO_2$, $O_x$) and made them unsuitable for future use. We caution end users to carefully select an appropriate sensor package given pilot information about the emission sources in their target site.

A growing body of literature highlights the potential value of LCS technologies for Sub-Saharan Africa and other low-resource settings (Subramanian and Garland, 2021; Wernecke and Wright, 2021; Rahal, 2020; Sewor et al., 2021; Awokola et al., 2020). We found that our LCS surface observations were consistent with the only other available data sources in this region (remote sensing data and model products) and data from similar studies across SSA. This suggests LCS have a key role to play in providing reliable information on general air quality conditions and trends in regions without a historical record. Advancements in machine learning techniques show how LCS can be used for source identification and attribution in regions where little quantitative information currently exists on dominant emission sources (Hagan et al., 2019; Thorson et al., 2019). While LCS in SSA show promise, many of the issues experienced in this study stemmed from a lack of in situ

reference monitors. Additional reference grade monitors throughout the region may help circumvent issues related to calibration modelling and quality assurance. A regional, shared facility would enable periodic, regionally representative collocations without requiring every country to establish its own regulatory network. Recent research has improved our ability to synthesize data from networks of LCS through computational calibration solutions which minimize the need to transport and collocate each individual monitor separately and increase the spatiotemporal resolution beyond that of

reference networks (Buehler et al., 2021; Malings et al., 2019a; Kelly et al., 2021; Considine et al., 2021; Sahu et al., 2021). Concurrently, policy-focused researchers are helping to bridge the gap between governments and AQ scientists by creating comprehensive frameworks which provide systematic procedures to establish regulatory AQ monitoring networks in regions without them (Gulia et al., 2020; Pinder et al., 2019). In the meantime, we found support from local universities, which helped maintain the pilot deployment of this LCS network. We expect that any AQ program in SSA will benefit from

building long-term, local capacity and knowledge transfer systems for training on-site staff and for receiving their feedback and guidance.

**Code availability**

The basic random forest hybrid and quadratic regression model codes are available in the supplemental information of the

890 original manuscript (doi:10.5281/zenodo.1482011). The k-nearest neighbor, high dimensional model representation, and multi-linear regression model codes are proprietary products of QuantAQ, Inc., contact David H. Hagan with inquiries.

**Data availability**

The dataset used in this analysis is available as an open-access Dryad depository (doi:10.5061/dryad.cz8w9gj4n). The

895 depository hosts pre-processed ARISense and reference datasets from the pre-deployment and post-deployment collocations, pre-processed RH-corrected OPC-N2 and MicroPEM datasets from the Malawi collocation, and collated ARISense datasets from the 1-year deployment at each of the three monitoring sites in Malawi. Please contact the corresponding author regarding raw data inquiries.

**Author contribution**

AG was responsible for conceptualization and funding acquisition. AG, EC, DH, and AB developed the methodology. EL, AG, and AB executed the deployment experiments. EC, DH, and AG provided supervision. DH and CM developed software. AB, EC, EL, and AG performed data analytics and visualization. AB wrote the original draft. CM, DH, EL, EC, and AG participated in review and editing.

**Competing interests**

Eben Cross and David Hagan are the co-founders of QuantAQ, a for-profit company which marketed the ARISense (since discontinued) and is actively developing and marketing sensor-based instrumentation.

## Acknowledgements

We would like to acknowledge funding from the National Science Foundation under Coupled-Natural Human Systems Award Number: 1617359. This work benefitted from Elliott Hall, who executed gravimetric filter analysis and contributed to computational analysis of the MicroPEM and OPC-N2 collocation data sets, and from Jillian McNaught through her contribution to the acquisition of the GIOVANNI data sets. Carl Malings would like to thank Naomi Zimmerman and the Carnegie Mellon University RAMPs Team for their assistance in developing low-cost sensor calibration approaches and acknowledge the EPA funding source under assistance agreement no. 83628601 and EPA Grant Number R836286, as well as the Heinz Endowment Fund Grants E2375 and E3145. He would also like to acknowledge his support by an appointment to the NASA Postdoctoral Program at the Goddard Space Flight Center, administered by USRA through a contract with NASA. This work benefited from State assistance managed by the National Research Agency under the "Programme d'Investissements d'Avenir" under the reference "ANR-18-MPGA-0011" ("Make our planet great again" initiative). Ashley Bittner would like to thank Ky Tanner for contributing to gravimetric filter analysis, Wyatt M. Champion for his contribution to Fig. 1, Nathan Williams (Carnegie Mellon University) for logistical support with ARISense repair, and all members of the Grieshop Atmosphere and Environment Lab. For their assistance in coordinating the collocation periods in North Carolina, we would like to thank the North Carolina Department of Environmental Quality and the U.S. Environmental Protection Agency and all dedicated employees including Sue Kimbrough (U.S. EPA), Richard Snow (U.S. EPA), Kay Roberts (NC-DEQ), Timothy Skelding (NC-DEQ), Joette Steger (NC-DEQ), and Vitaly Karpusenko (NC-DEQ). Finally, we would like to thank all project principal investigators including Dr. Pamela Jagger, Dr. Thabbie Chilongo, Dr. Charles Jumbe, Dr. Rob Bailis, Dr. Jason West, and Dr. Adrian Ghilardi, principal interpreter and field work assistant Twapa Ghambi, equipment assistants Dominic Raphael and Misheck Mtaya, and all study participants from the villages of Mikundi and Makaula in Mulanje, Malawi.

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
