# Peer review of "Performance Characterization of Low-cost Air Quality Sensors for Off-grid Deployment in Rural Malawi"

_Atmospheric Measurement Techniques, 2021_

## Referee Comment (RC2)

[referee-annotated manuscript omitted]

---

## Author Comment (AC1)

We would like to thank both reviewers for their helpful feedback. The revisions involve clarifications to the text, reordering of the sections to improve flow, and changes to the way the data are presented in Figures 1, 2 and 6.

Below we respond to the reviewer #1's comments in detail. Reviewer comments are in black text while author responses are in **bold purple text.** Excerpts from the original manuscript are shown in *italicized purple text.* Revisions to the manuscript are shown in *italicized red text.*
* * *
**Response to Reviewer 1**

This article discusses the calibration via colocation experiments and five different calibration models, deployment, stability, and recolocation/post deployment check of low-cost air quality sensors (specifically ARISENSE measuring PM, O3, NOx, and CO) in Malawi. Low cost sensors are increasing in use, specifically in areas with less infrastructure or resources for reference air quality monitoring stations, and long-term deployments are useful to see how they perform. Comparing sensor calibration robustness over time is also useful. I recommend eventual publication. However, there are a few large issues with this paper that must be addressed first. General comments first, then specific comments.

General Comments

This paper feels like an add-on/afterthought to another paper (on the air quality findings by the low-cost sensors in Malawi, which is mentioned by the authors as a separate publication in prep). That's an efficient data use strategy, but means that this study was not designed to optimize it's stated main goal of sensor calibration scheme comparison and robustness. There are no reference monitors available for comparison during deployment, so the ability of the scientists to really understand how their calibrations performed over time and how performance changed when exposed to different conditions, beyond knowing when the sensors are returning non-physical data (e.g., negative values), is unclear.

> **We did not intend to portray the main goal of this paper as an optimized sensor calibration scheme comparison. Our overarching goal is to get data in never-studied regions using affordable LCS technology. The main goal of this paper is to fully address the methodology of deploying this specific technology in our target deployment site and to provide examples for approaches others may use in the situation where no *in situ* reference monitors are available. More plainly, our goal was to determine and share how to use and infer results from the equipment we deployed.**
> **To clearly establish the goal of paper, we added the following statement in the Introduction:**

> "*Our overarching goal was to assess the viability of establishing and maintaining a small, temporary network of LCS monitors in Malawi until a more formal governmental regulatory monitoring system can be established. Given that comparison to regulatory grade equipment in Malawi was not possible, the objective of this work was to devise an*

*alternative methodology to evaluate the ARISense technology for accuracy, precision, and stability over the 1-year pilot deployment.*"

**Further, to establish this papers' independence from the forthcoming paper, we reframed how the objectives were both defined (Section 1) and addressed (Section 4). In the Conclusions (Section 4), we comment on the viability of using such a technology for our goal and provide insight on how to improve future efforts.**

➢ **To improve readability, we also reorganized sections to improve the flow of the paper. The last paragraph of Section 1 was rewritten as:**

"*In Section 2.3 and 2.4, we discuss separate collocations of the gas sensors (in North Carolina, USA) and particle sensor (in Mulanje, Malawi) with reference or semi-reference instruments (Section 2.2). We use the collocation data and quantitative assessment metrics (Section 2.5) to compare the performance of five modelling approaches to calibrate the gas sensors (Section 3.1) and estimate error in the particle sensor data (Section 3.2). After deployment to Malawi (Section 2.6), we qualitatively assess how the ARISense performed in the field using contextual information about nearby emission sources, diurnal trend data, and an inter-comparison of calibrated gas model observations (Section 3.3 and 3.4). In Section 3.5 and 3.6, we compare the results to remote sensing and reanalysis data products and surface measurements from similar environments in SSA. Finally, in Section 3.7, we qualitatively assess the long-term stability of the sensor readings and calibration models in Malawi by comparing seasonally similar ambient data collected 1-year apart at the same location. In concluding (Section 4), we use evidence from this pilot study to characterize the benefits, limitations, and robustness of this technology and methodology for our application: collecting AQ data in under-studied regions. Additionally, we offer guidance on considerations to improve future remote deployment efforts.*"

➢ **We realize the approach we had to take to calibrate and assess the data was not ideal, but that was inevitable, given the lack of reference monitors in the region. We believe we have reframed the paper to more clearly show why we took this path and to clearly state when we are using quantitative vs. qualitative inference. Finally, we believe we sufficiently acknowledge the shortcomings of our methodology, and as a result, we express to the extent possible the uncertainty associated with quantitative values.**

A comparison of sensor data to satellite data is done; however, the authors themselves acknowledge past work that shows that, in Africa, comparison of satellite to ground data is non-ideal.

➢ **For clarity, we changed the name of Section 2.7 from "***Remote sensing data***" to "***Remote sensing and reanalysis data***". MERRA-2 is a "reanalysis" data product, so it is the output of an atmospheric chemistry model that has "assimilated" other data (including but not limited to satellite data) in its estimations.**
➢ **We believe this comparison is adding to the literature on surface-to-satellite comparisons over Africa. More broadly, comparison of satellite data to single-point ground measurements is never "ideal" but it is done all the time. We believe this comparison adds value to the discussion and literature on such comparisons. For example, this information may help inform decisions about where and why**

**permanent ground-stations should be added in Africa. We added a new opening to the first paragraph of this section to emphasize our purpose for this comparison and the main outcome:**

**"***In the absence of in-situ surface data, we rely on satellites and models to estimate surface air quality. To contribute to the literature on surface-to-satellite comparisons over Africa, we compared calibrated ARISense CO observations to a satellite observation (MOPITT) and a model estimate (MERRA-2) for the study region. We confirmed that all three data sets reported similar annual qualitative trends, although they disagreed in magnitude.***"**

➢ **In the absence of "ideal" data from well-maintained regulatory-grade surface monitors, we are using what we can (low-cost sensor data), while being honest about the limitations.  In the absence of any in-situ data, we only have satellites and models; this analysis compared with both and confirmed they are all seeing the same trends, although they disagree on magnitudes.**

➢ **We believe that we have correctly pointed out the limitations of this comparisons and shown the qualitative agreements, which are useful. We have added to the text in Section 3.3 "Comparison of ARISense CO to remote sensing data" to make this clearer for the reader:**

*"Monthly mean CO ARISense values were 2 to 4 times higher than those reported by MOPITT and MERRA-2. We found differences of 175 to 200% between the annual mean CO concentration from ARISense and MOPITT, depending on the site, and even larger differences (up to 360%) with MERRA-2. Differences between MOPITT and MERRA-2 were smaller (30 to 35%). There are few comparable studies available to explain these differences, which are greater than previously reported in the literature available for SSA. One study in South Africa reported relative differences of ±40% between ground-based CO measurements and Aura satellite observations at Cape Point station (Toihir et al., 2015). Many studies found good agreement (within 10-20% bias) between ground measurements and MOPITT observations, but this was for Total Column CO, and the observations were not limited to comparisons over Africa (Buchholz et al., 2017; Emmons et al., 2009, 2004; Yurganov et al., 2008, 2010). However, these studies found negative satellite bias when intense biomass plumes affected observations, when CO levels were low in the Southern Hemisphere, or when atmospheric CO levels changed rapidly (Buchholz et al., 2017; Emmons et al., 2004; Yurganov et al., 2008, 2010). Each of these conditions could reasonably be expected to occur in the southern Africa troposphere, potentially explaining differences observed between the ARISense and remote sensing observations in this study.*

*This comparison of low-cost sensor surface data, satellite observations, and model estimates in Malawi suggests each of these resources can give consistent information on qualitative, long-term trends in a region without ground-based reference monitoring. However, because of inherent differences in spatial and temporal resolution, each observation will likely disagree in magnitude. Satellite retrievals and real-time surface measurements do not result in directly comparable quantities. Satellite data are generally collected as a once-daily flyover observation, averaged over a ~12,000 square kilometer area (corresponding to 1° spatial resolution). In contrast, the ARISense data were 1 min resolution, fixed-site, long-term point measurements at the surface. Further, the ARISense data were collected near visually identified biomass emission sources and were not representative of background conditions. Meanwhile, the satellite observations provide an*

*estimate of regional background conditions. Despite these differences, the MOPITT, MERRA-2 and ARISense data sets agreed on the long-term seasonal trends present in this region, and even corroborated site-to-site differences (e.g., higher mean CO at University compared to Village Mean site). These findings suggest the ARISense captured synoptic-scale variation in CO, but comparison to remote sensing data does not allow for a quantitative assessment of data collected at higher temporal resolutions."*

There are mentions of 'known emissions' near each site, but no clear definition and numbering of these emissions or discussion of their distance/location from the site.

- ➤ **We added markers to indicate the location of visually identified biomass cookstoves in Fig. S12 and S13 (see Supplement .pdf attached to this response).**
- ➤ **We also added the following context to Section 2.6:**

  *"ARI013 ("Village 2" site) and ARI014 ("Village 1" site) were deployed < 5 km apart in two rural residential villages in Mulanje, Malawi, adjacent to many households. Almost all rural households in Malawi (99.7%) use solid fuels (e.g., firewood, charcoal) for cooking (National Statistics Office, 2017). Emissions from widespread biomass cookstove use are known to impact local ambient air quality (Aung et al., 2016; Zhou et al., 2011; Amegah and Agyei-Mensah, 2017). Homes regularly using biomass cookstoves within 50 m of the monitoring sites were visually identified at the onset of the study (shown as red cross-hatches on Fig. S12 and S13)."*

If the goal of the study was really to compare long-term performances of these models, this study was not ideal for that.

- ➤ **We agree with the reviewer that this study is not ideal for that goal, and this is one of several objectives of the study. We did not intend to portray this paper as a comparison of the long-term performance of the calibration models. The goal of the study is to determine if measurements made during the 1-year deployment were reliable and consistent. Characterizing and comparing the long-term performance of the models was a method used to support that goal. We hope that we resolved this issue with changes made in our first response statement above.**

Perhaps this could still be done in part—could you also pretend the initial colocation was a full experiment, .

- ➤ **We used the initial colocation data in N.C. to develop and quantitatively assess the performance of the models before deployment to Malawi. We cross-validated the models by separating our full data set into separate testing and training data sets and evaluating the performance using EPA-recommended assessment metrics.**
- ➤ **Because of the lack of reference monitors in Malawi, we were unable to quantitatively assess model performance during the deployment to Malawi. We qualitatively assessed the performance by comparing the diurnal trends predicted by each model.**
- ➤ **We cannot use the post collocation data collected in N.C. to quantitatively assess the long-term performance of the models, because the sensors were degraded during the high-concentration biomass burning experiments conducted in Malawi after the 1-year deployment. Unfortunately, we cannot assess long term model**

**performance in any other way than our 1-year in situ comparison discussed in Sect. 3.7.**

➢ **We believe we sufficiently address the limitations and shortcomings of our approach in the final three paragraphs of Section 3.7. To make this clearer to the reader, we added text in Section 3.7:**

"*Ultimately, the $O_x$, NO, $NO_2$ sensors were permanently altered by the biomass burning emission experiments in Malawi, leading to the poor performance during post-deployment collocations with reference instruments in NC. Given these dramatic changes in sensor responses, the models were unable to generate reasonable concentration values from sensor signals and consequently, we were unable to use the post-deployment collocation data set to quantitatively assess long-term model performance.*"

I acknowledge that this is hard in this particular region; however, another major issue is that there is no one from the region involved as an author on this study.

➢ **We would like to clarify that although no one from the region is listed as an author on this specific manuscript, the larger study involved scientists and civilians from the study region. We do not wish to under-emphasize the role that our local collaborators played and so we mention, by name, the many persons that contributed to this work in the Acknowledgements section.**

➢ **For clarity, we explain how each author contributed to the manuscript in the 'Author contribution' section of the manuscript. In conferring authorship, we adhered to the criteria given in the Vancouver recommendation (https://i.ntnu.no/wiki/-/wiki/English/Co-authorship):**

*"The person in question must have made a substantial contribution to the conception or design of the work; or to the acquisition, analysis, or interpretation of data for the work.*

*1. She or he must have been involved in drafting the work or revising it critically for important intellectual content.*
*2. She or he must have approved the version of the manuscript to be published.*
*3. She or he must agree to be accountable for all aspects of the work in ensuring that questions related to the accuracy or integrity of any part of the work are appropriately investigated and resolved. In addition to being accountable for the parts of the work he or she has done, an author should be able to identify which co-authors are responsible for specific other parts of the work."*

➢ **Our local contacts at the University of Lilongwe Centre for Agricultural Research (CARD) are in a different research discipline and as such were not considered to have contributed in the ways outlined above in the development of this paper. We attempted and failed to identify and involve interested students at CARD, possibly because air quality science is not included in/or a focus of the CARD curriculum. A few years into this project, we established a collaboration with, and eventually a monitoring site at, an engineering-oriented university in Malawi, recently renamed the Malawi University of Business and Applied Sciences (MUBAS), but they were not involved with collecting or interpreting any of the data presented here. We will engage them further in the analysis and documentation of the full data set. Given the content of this initial paper in the two-paper series,**

**we decided to limit authorship to those involved in the establishment of the sampling sites (AG, EL, and AB), developers of the technology (EC, DH) and developers of the calibration tools used (CM, DH, EC). Our unique contributions are further described in the "Author Contribution" section at the end of the manuscript.**

Involving local scientists would have greatly increased the scientific merit of this paper in a few ways.

➢ **We agree and we plan to engage with researchers at MUBAS and invite them to participate in the authoring of the second paper in this two-paper series.**

1. Better understanding of conditions on the ground and local context

➢ **Three authors (AG, EL and AB) identified and established the monitoring sites through on-the-ground visitations and discussions with local residents. AG and AB maintained contact with local staff throughout the duration of the study, listened to their comments and concerns, and passed information on the equipment and progress of the study to staff and project participants. AG and AB annually visited the sites.**

➢ **At the onset of the study in July 2017, AG, EL, AB and CARD faculty participated in village-wide meetings at Village 1 and Village 2, attended by the village chiefs, their families, and many residents. Residents shared their comments and questions, and some volunteered to be study participants. During this meeting, Dominic, a resident in Village 2 (mentioned in the Acknowledgements) expressed his interest in helping our team execute the study. He was hired as our interpreter, guide, person of contact, and assistant for the remainder of the study (July 2017 to March 2021).**

2. Better data capture by regular maintenance

➢ **We hired two regular maintenance staff for the duration of the study: One at the Village 2 site (Dominic Raphael) and one at the University site (Misheck Mtaya). We originally identified and hired an additional maintenance assistant at the Village 1 site but failed to retain them. After their resignation, our Village 2 assistant took on the role of maintenance assistant for both Village sites.**

➢ **In all cases, field staff had limited technical background, and so maintenance (of which there is relatively little for these sensor units) was not a major role. Instead, they assisted with data download, checking on continued operation, and with some troubleshooting via remote communication with us when the monitors failed to operate.**

3. Better understanding of how these sensors might be used

➢ **The target user for this technology is a government or private institution. The technology is not intended for citizen science purposes. We held a collaborative planning meeting with government officials in July 2017 in Lilongwe, Malawi before the beginning of the study. Another collaborative, reflection workshop was planned in summer 2020 to mark the conclusion of the study, but it has been postponed due to the on-going COVID-19 pandemic.**

4. Better understanding of how well corrections factors on data might be applied. (e.g., are the correction factors easy to apply with limited computing power and limited software?).

> **In the development of this study, it was determined that the calibration of the sensor packages would be the combined responsibility of the product developer and researchers. The end user (e.g., a national research institution or governing body) was not expected to be involved during that stage of product development. Therefore, we did not consider the ease of application or computing power/software limitations.**
> **While during this study, the post-processing of data was required (developing and validating the models was a key component of this work), the future application of these models/calibration factors in real-time is feasible. The models used are light-weight, pre-trained models that run server-side and are thus not limited by computational power on the edge device itself.**

Understanding the actual use, by people in the region, of these sensors could have made up for some of the lack of reference data comparisons by discussing another essential facet of low-cost sensor use (the people using them).

> **We agree with the reviewer that documenting use cases/how these sensor packages will be used would be a great next step. However, this will require integration of efforts with national or regional-scale regulatory or research institutions. We have continued these conversations with representatives of such organizations in Malawi (e.g., via our link with MUBAS) but this is a longer-term effort and is outside the scope of this focused paper.**

Specific comments

Table 1: define QR in table caption

> **Original caption: "**Description of the five calibration modelling approaches and data inputs for each gas sensor and model combination (CO = carbon monoxide, NO = nitrogen oxide, $NO_2$ = nitrogen dioxide, $O_3$ = ozone).**"**
> **Revised caption: "Calibration modelling inputs for each gas sensor (CO = carbon monoxide, NO = nitrogen oxide, $NO_2$ = nitrogen dioxide, $O_x$ = oxidants) and model combination ('All' indicates k-nearest neighbor (kNN) hybrid, random forest (RF) hybrid, high-dimensional model representation (HDMR), quadratic regression (QR), and multi-linear regression (MLR)."**

How reasonable is developing individual models for each sensor and each gas component?

> **The different gas sensor types (CO, NO, NO2 and Ox) require separate and individual calibration approaches because the physical basis of their response and cross-sensitivity to environmental conditions can and does vary.**
> **The different sensor packages (ARI013, ARI014, and ARI015) could potentially be calibrated by a 'group' model. However, inter-unit variation among the sensor packages (Figure S3) motivated the use of individual models. Further, we determined that the small size of this monitoring network (N=3) made this option feasible. Other studies have investigated the tradeoffs of different approaches using larger networks (Malings et al., 2019).**

> **In any case, we posit that it is reasonable and common to develop and use calibration factors unique to an analytical instrument. In the case of the U.S. Federal Reference Methods, it is a requirement.**

60 minute averaged data: will miss 'events' potentially like cooking/agricultural and trash burning—can you speak to the significance of these types of events?

> **We agree with the reviewer that 60-min averaged data is unsuitable to visualize trends in short/variable emission events like trash burning. We acknowledge that nearby emission events can impact the representativeness of ambient, background measurements and we feel we communicate this throughout the paper. We plan to quantitatively explore the impact that source emissions have on background levels in the second paper of this two-paper series. However, we clearly see the influences of village- and household-scale cooking activity, even at 60-min averaging intervals, at least in the aggregated CO and $PM_{2.5}$ diurnal trend data (Fig. 3 and Fig. 7) for the villages.**

Figure 1: I found this confusing. And hard to tell if the different models worked better or worse on the different sensors, or there was consistent agreement from looking. Are they the same graph just color-coded differently? If so you could make sure no overlaps in color on the two graphs. I think you are trying to convey a bit too much information on one graph for ease of interpretation. A table, or clearer labeling with different colors, or an extra panel, might work better to convey your message, which is a bit lost right now. I think a table (like those in the supplementary nfo) or adding some numbers of relative performance in your text could make the determination of which model worked best clearer for the reader.

> **Based on the reviewer's feedback, we decided to change the way information was presented in Figure 1 (revised Fig. 1 below). We removed panel (b) and used four panels, one for each gas sensor. We selected new color-blind friendly colors.**
> **Based on Reviewer #2's feedback, we decided to use the EPA-recommended RMSE metric to assess error, rather than MAE, which was previously shown on the y-axis of Figure 1. The RMSE and MAE values were similar, however, and did not change the interpretation of results or the conclusions of the paper.**
> **To emphasize that the kNN model performed the best for all gas sensors, we used a darker color and increased the marker size and the border weight compared to the markers used to represent the other model types.**
> **We also added the following discussion about the relative performance of the models in this section of the results:**

*"In almost all cases, the kNN hybrid model returned higher $R^2$ values and lower RMSE values than any other model. The RF hybrid model attained similar, and occasionally higher $R^2$ values than the kNN hybrid, but it had higher (and therefore worse) RMSE values by comparison. Further, the kNN hybrid model showed the least inter-monitor variation in performance. In Fig. 1b-d, the kNN hybrid points are closely clustered together, suggesting that this model was able to attain approximately the same performance for each of the three ARISense. Conversely, the other models, in particular the RF hybrid and MLR, showed a wide range in performance across the three ARISense. Even if another model was able to attain performance metrics higher than the kNN hybrid (e.g., HDMR and MLR CO models in Fig. 1a) it was only for one of the three ARISense monitors, never all three. Given that we seek an approach uniformly applicable to all the gas sensors and*

*all three ARISense, any model other than the kNN hybrid was unsuitable. Additionally, the MLR failed to meet target values for some ARISense-gas sensor combinations (Fig. 1a-b)."*

➤ **We also plan to add a summary table of the performance metrics for all models to the supplementary information for the revised submission.**

[Figure]

**Figure 1 (revised):** Performance comparison of gas sensors (a) CO, (b) NO, (c) $NO_2$, and (d) $O_x$ as calibrated by the five types of calibration models adopted for this study (kNN hybrid, RF hybrid, HDMR, MLR, QR). The model type is indicated by color and marker shape. An individual data point represents the paired metrics (MAE and $R^2$) for one ARISense monitor. Since there are three ARISense (ARI013, ARI014, ARI015) monitors, there are three markers for each gas sensor-model combination. MAE is mean absolute error. $R^2$ is the coefficient of determination (-infinity $\leq R^2 \leq$ 1). The lower left corner region of each panel indicates the highest performance based on these metrics.

For the atmospheric pressure versus sensor performance issues: are there any studies you could quickly compare your work to? Maybe some done in Boulder, for extrapolation? If not, I don't think that limits this work at all, as the difference in elevation is not huge.

➢ **We added an additional reference to the literature to strengthen our argument:** "*Further, others have shown no statistically significant change in electrochemical sensor sensitivity due to changes in pressure (Popoola et al., 2016).*"

Figure 2: These plots are somewhat confusing and hard to interpret, in my opinion. I would at least add a color scale bar. I think there's likely an easier way to display this info. Also I searched your text and I don't see V1,V2, Uni defined anywhere? I assume it is village 1 village 2 university, but I would repeat that info in the graph caption and maybe in the text when introducing the sites define that V1=village 1 so readers just browsing the graphs can figure out the meaning more quickly.

➢ **We removed all instances of V1, V2, and Uni and only use 'Village 1', 'Vilage 2' and 'University' throughout the manuscript. However, to aid in interpreting this figure, we changed "NC" to "Colocation", and each deployment site to "Field" to emphasize the point of this figure. Displaying the colocation and field data as bivariate histograms allows us to visualize regimes where our models are poorly constrained (i.e., regions where the colocation and field 'blobs' do not overlap). We added additional text to the caption (shown in red below) for clarification. We plan to choose a color blind friendly color scheme to denote the three sites in the revised manuscript.**
➢ **The colorscale denotes the number of data points in that location, or the "density". Because of data gaps throughout the field data, the quantitative density values are different for each panel, therefore a single colorscale bar would not accurately represent all panels. Further, we think that adding two colorscale bars (colocation and field) for each panel (a-i) would only further complicate this figure, especially since the actual density values are not relevant to our message. We only want to communicate data density qualitatively, and we feel the explanation in the caption conveys this sufficiently to the reader: "***Density is reflected in the color scheme; Darker colors indicate more data points in that region***".**
➢ **We display this information using simpler, traditional histograms in the supplementary information, but that presentation prevents us from identifying important interactions between the variables (e.g., $O_x$ voltage and T) that might impact model performance. To further emphasize the focus of this figure, we added markers ('x' and 'o') to identify the regions discussed extensively in the text (revised figure below). This allows the reader to synthesize information between the text and figure and find the exact region that we are referring to more easily. We added to the text in the section (shown in red) to point the reader to the exact region we are referring to:**

*"Figure 2e shows the maximum ARI014 CO differential voltage in Malawi (350 mV) was 3 times higher than the maximum voltage registered in NC (100 mV). This high CO regime is denoted by an 'x' on Fig 2e. This difference was aligned with observations of nearby sources (Fig. S12 and S13). We expected higher CO in Malawi than in NC, where biomass burning is less common and emissions from other sources (e.g., vehicles) are controlled by strict federal regulation. ARI014 was deployed in a densely populated village, adjacent to more biomass cookstove activity than ARI013 or ARI015 (Fig. S13).*

*The O$_x$ differential voltage ranges were the most dissimilar between the collocation and deployment environments* (denoted by a '+' on Fig. 2a-c). *The most frequent regimes, the heaviest shaded regions in Fig. 2a-c, did not overlap for any of the ARISense. In NC, the relationship between the O$_x$ sensor voltage and ambient temperature was positive and approximately monotonic. Generally, higher temperatures facilitate ozone production, therefore this relationship fit our expectation for an urban site in a single season. However, the positive relationship between O$_x$ sensor voltage and temperature did not always hold in the deployment sites. Figure 2a-c shows a high temperature-low ozone regime in Malawi that was not present in the NC data* (region denoted by a '+' marker)."

[Figure]

**Figure 2 (revised):** Bivariate distributions of gas sensor calibration model data inputs (RH, T, and O$_x$, CO, NO, and NO$_2$ differential voltage) for each ARISense monitor using kernel density estimation. Density is reflected in the color scheme; Darker colors indicate more data points in that region. Training data collected during collocation in North Carolina are shown in grey; data collected during field deployment to Malawi are shown in color. Regions where the field deployment distributions overlap with the N.C. collocation distributions indicate the regimes for which the calibration models were trained. Regions where the deployment location distributions extend beyond the N.C. collocation distributions indicate regimes where the calibration models extrapolated to estimate pollutant concentrations. These regions are indicated by overlaid markers 'x' and '+'.

Line 279 : would recommend you to define more what you mean by 'well controlled'

➢ **This statement was rephrased as: "***We expected higher CO in Malawi than in NC, where biomass burning is less common and emissions from other sources (e.g., vehicles) are controlled by strict federal regulation."*

Line 288 can you expound a bit on the 'difference in ozone precursor regimes?' I know a separate study is likely coming out about the measurements themselves, and this will be expounded on, but this line as it is just is hanging there begging for a bit more info.

➢ **We agree with the reviewer and in response we decided to remove discussion of this topic. It is not necessary to understand the main point (that we lack representative collocation data, particularly for the oxidant sensor).**
➢ **We plan to address this topic in the second paper of the two-paper series, where we can do a first-order estimate of ozone precursor regimes at the collocation and deployment sites (using O3 to NOx ratios).**

Line 295 discussions: is calculating a diurnal trend really a way to see if the models are transferable, if you don't have any nearby ground-based air quality data? How did you get your local knowledge of air quality? (anything to cite?)

➢ **We posit that calculating the diurnal trends is a qualitative approach to assess if the models are transferable, since the deployment site does not have ground-based air quality data for quantitative comparison.**
➢ **The diurnal data show we're in the right ballpark in accordance with what we know about atmospheric chemistry (based on our expertise as air quality scientists), these specific deployment sites (based on our firsthand observations of diurnally variable emission sources), consistency between combustion tracer diurnal trends (CO and PM), and what other studies in Africa have shown. While this doesn't necessarily show the models are transferable perfectly, it does suggest they likely capture some of the major features of diurnal trends at these sites.**
➢ **To better support our approach in this section, we rephrased the text and added references:**

*"Since the deployment site does not have reference data for quantitative comparison, we calculated and compared the annual mean diurnal trends of each pollutant, at each site, as predicted by the five models to qualitatively assess the transferability of the calibration models to Malawi. Our definition of a transferable model required that it produce: (a) non-negative concentration values and (b) diurnal trends consistent with our first-hand observations of nearby emission sources and their timing, our knowledge of the ambient trends in regions with widespread biomass cookstove use (Dionisio et al., 2010; McFarlane et al., 2021; Subramanian et al., 2020) and atmospheric chemistry."*

*"This analysis can contribute to our confidence in the estimated concentration values and trends, but ultimately cannot address or estimate the quantitative error."*

Line 319: sensor has less RH/T interference at higher concentration—can you provide a reference for this? It makes some logical sense but would be nice to expand this thought.

> **This statement was rephrased to:**
>
> *"RH/T interference induced on the CO-B4 sensor, ~0.2 mV/ppb (Lewis et al., 2016) has relatively less influence on overall sensor response in the higher voltage (i.e., concentration) regime."*

Also, are there any chemical species co-emitted with CO that, at higher levels of CO, would also influence CO?

> **For the major emission sources near the monitoring sites (biomass cookstoves), CO is co-emitted with fine particulate matter (PM). We do not expect co-emitted PM to influence the formation of CO (chemically). Photochemical oxidation of VOCs (also co-emitted during biomass burning) can form CO, but this is on much longer time scales (hours-days), so we expect this to be a background contribution and not one that influences diurnal trends in the way that local emissions do.**
> **But it is possible, and at extremely high concentrations probable, that co-emitted pollutants (e.g., gas phase hydrocarbons, other reactive gases, fine particulates, etc.) can influence the response of the CO sensor due to cross-sensitivity. For example, ethylene oxide and carbon monoxide sensors share the same catalyst and structure (but have different pre-filters), therefore small VOCs could interfere with CO sensor performance. At the low ambient concentrations in Malawi, we do not expect this to be an issue because VOC concentrations are typically 2-5 orders of magnitude lower than CO in biomass burning emissions (Akagi et al., 2011) and the cross sensitivity of the CO-B4 sensor to an example hydrocarbon (acetylene) is less than 1% (https://www.alphasense.com/wp-content/uploads/2019/09/CO-B4.pdf).**
> **However, in wildfire-like conditions, or during the high-concentration biomass burning fenceline experiments we conducted after the ambient deployments, certain sensors (even high-quality reference instruments like the 2BTech 202 Ozone analyzer) can be rendered essentially useless due to cross-sensitivities** (Long et al., 2021)**.**

How did you characterize the known CO sources?

> **We visually identified nearby biomass cookstoves through on-the-ground site walkthroughs and by spending full days in the village sites over multiple weeks.**
> **We characterized a sample of the known CO sources visually and with emission measurement equipment (Section 10 of SI - this is a separate paper and research study which will be published later).**

The grid averaged for CO surface concentration is 12,000 km2, right?. How many different CO sources are within an area of that size that would cause heterogenous CO measurements?

> **The spatial resolution is 1° (corresponding to ~12,000 km$^2$) for the MOPITT CO surface concentration (Table S5) and 0.5° x 0.625 for the MERRA-2 product (Table S6). Every combustion site is a potential CO source, and there are quite a lot of them contained in each grid cell at those spatial scales. We do not know how many different CO sources are within each grid cell nor do we try to estimate it.**

**Further, we postulate that the number of CO sources is important, but so is the relative proximity of the point measurement to the source. The ARISense were placed in the middle of a cluster of CO sources (biomass cookstoves in the village) while the satellite is seeing sources from those two villages, plus many other villages, plus the less-populated areas between them that make up the majority of the land cover.**

➤ **We recognize that estimates from a satellite observation and measurements collected by a single surface instrument are not directly comparable quantities. We emphasize that we do not expect them to perfectly agree, since they are not measuring the same thing, but we can still draw qualitative conclusions by comparing them. And although an area-average and a point value are not directly comparable, they are the only measures available for comparison in this region, so when making this comparison, we offer educated guesses about the reasons for observed differences (i.e., the proximity of the ARISense to a cluster of CO sources – the biomass cookstoves in the villages).**

Figure 4: Label individual village sites, be consistent with naming of your sites.

➤ **Data from the individual village sites are not shown in this figure. Data from the individual village sites was averaged in the data points shown. To make this point clearer to the reader, the term "Villages" was renamed to "Village Mean" (defined in Section 2.7).**

➤ **Previously in Fig. 4, the 'Village Mean' color was identical to the Village 1 color; this was misleading the reader. Now, a different color scheme has been selected to differentiate the Village 1, Village 2, and Village Mean data. Additionally, the naming convention now appears consistently throughout the paper (i.e., Village 1, Village 2, and Village Mean).**

Line 651: I assume the 'relatively short' and '100 hours' is 100 hours for the whole year? Or is that per biomass burning episode?

➤ **Approximately 100 hours was the total duration of all biomass burning emission experiments conducted at the conclusion of the 1-year period of ambient sampling.**

➤ **Original: "Operation beyond specified conditions, combined with repeated, although relatively short (< 100 hours), exposure to high concentration gases during the post-deployment emissions monitoring experiments, made the three less robust sensors unsuitable for future use."**

➤ **Revised: "Operation beyond specified conditions, combined with ~100 hours of exposure to high concentration gases during the post-deployment emissions monitoring experiments, apparently damaged the three less robust sensors (NO, NO$_2$, O$_x$) and made them unsuitable for future use."**

Also, are you certain that O3, NO, and NO2 came from 'fresh' biomass burning emissions, and not other sources (e.g., NOx from diesel trucks with poor emission controls)? O3 from biomass burning not so straightforward and a result of aging of biomass burning emissions.

➤ **We did not sample near diesel vehicles. We agree that, as a secondary pollutant, O$_3$ was not emitted from the 'fresh' (primary) biomass burning emissions.**

➤ **The O$_x$, NO, and NO$_2$ sensors were likely indirectly affected by high concentrations of PM and volatile gases (hydrocarbons, formaldehyde, etc.) emitted during the**

**biomass burning experiments, not necessarily high concentrations of the target species (O₃, NOₓ) themselves. Very high concentrations of emissions can chemically degrade or contaminate the sensors, for example, the catalyst or electrolyte can be affected or depleted by repeated interactions with high concentrations of non-target species emissions. Further, if there are high concentrations of fine PM permeating the inlet and flow line, it can condense and block or attenuate the sample flow rate.**

➢ **We made this point clearer to the reader by adding more explicit text in Section 3.7:**

*"These ambient sensors (except for possibly the CO sensor) were likely affected by high concentrations of PM and volatile gases (e.g., hydrocarbons, formaldehyde, etc.) co-emitted during the biomass burning experiments. Very high concentrations of emissions can chemically degrade or contaminate the sensors, for example, the catalyst or electrolyte can be affected or depleted by repeated interactions with high concentrations of non-target species emissions."*

Would be helpful to have a map of the stations & a figure showing when sensors were collocated with and when the instruments weren't collecting data. There are some maps in the supplement, but a nice and simplified summary graphic in the main paper would be good.

➢ **We agree with reviewer and propose adding an additional Figure (Figure 0 below), which would become the 'new' Figure 1 in the revised manuscript. This figure was Fig. S10 in the Supplementary information, however Reviewer #2 suggested we move this map to the main manuscript. Further, we provide a timeline of the project, indicating which ARISense were deployed to which sites, and when they were collocated, deployed to the field, and used for an emissions monitoring campaign.**
➢ **We plan to add timeseries of the temperature data from the full year, example given below (Fig. S0), to show when the instruments weren't collecting data.**

[Figure]

| | May - June 2017 | July 2017 - July 2018 | July - Aug 2018 | Aug 2018 - Mar 2019 |
|---|---|---|---|---|
| **ARI013** | Colocation (N.C., USA) | Field deployment (Village 2) | Emissions monitoring (Village 2) | Colocation (N.C., USA) |
| **ARI014** | Colocation (N.C., USA) | Field deployment (Village 1) | Emissions monitoring (Village 2) | Colocation (N.C., USA) |
| **ARI015** | Colocation (N.C., USA) | Field deployment (University) | Emissions monitoring (Village 2) | Colocation (N.C., USA) |
| **ARI023 (OPC-N2)** | n/a | n/a | Colocation w/ MicroPEM (Village 2) | n/a |

**Figure 0 (previously Fig. S10):** Satellite map of Malawi (top), blue markers indicate ARISense monitoring sites and project timeline (bottom). The timeline indicates the activity conducted for each time period, and the location of that activity is given in parenthesis.  Image source: Google Earth Pro Version 7.3.4.8248. University, Village 1, and Village 2, Malawi, Southeastern Africa.  Borders and labels layer. Accessed: June 5, 2020.

[Figure]

**Figure S0:** Timeseries of temperature data from ARI015 (top), ARI014 (middle), and ARI013 (bottom) from the full 1-year pilot deployment in Malawi. Gaps in the timeseries indicate periods when the ARISense were not collecting data. Text labels indicate the causes of data loss: 'solar not keeping up' refers to insufficient solar power in the winter months; 'logging issues and unrest' refer to the combination of corrupted USB devices which failed to log data, and a period of social unrest in the southern region of the country which created unsafe conditions for our assistant to visit the monitors; 'collaborator visits for reset' indicate when a collaborator visited the village locations to replace the USB devices and update the firmware.

If possible, a map with direction of known emitters could help (would assist with the discussion in change in sensor performance with wind direction). As it is, emission sources are mentioned but are unclear where and what they are, and how information that they even existed was obtained.

> ➤ **We revised Fig. S12 and S13 (shown below) to show the known emitters.**
> ➤ **We added the following context (shown in red) to Section 2.6:**

"*ARI013 ("Village 2" site) and ARI014 ("Village 1" site) were deployed < 5 km apart in two rural residential villages in Mulanje, Malawi, adjacent to many households. Almost all rural households in Malawi (99.7%) use solid fuels (e.g., firewood, charcoal) for cooking (National Statistics Office, 2017). Emissions from widespread biomass cookstove use are known to impact local ambient air quality (Aung et al., 2016; Zhou et al., 2011; Amegah and Agyei-Mensah, 2017). Homes regularly using biomass cookstoves within 50 m of the monitoring sites were visually identified at the onset of the study (shown as red cross-hatches on Fig. S12 and S13).*"

[Figure]

**Figure S3:** Satellite image of Village 2 (1000ft scale), blue markers indicate ARISense monitoring sites (ARI013), red crosshatches indicate nearby biomass cookstoves. In addition to the explicitly marked cookstoves, each white square is a household, likely with its own cookstove. ARI013 was deployed to the Village 2 site and was mounted on the roof of the residence of the village chief (4 m above ground) in the Mikundi village of Mulanje, Malawi for 382 days from 6 July 2017 to 23 July 2018. Image source: Google Earth Pro Version 7.3.4.8248. Mikundi village, Mulanje, Malawi. 36.056°S, 35.535°E, eye elevation 626 m. Borders and labels layer. Accessed: June 5, 2020.

[Figure]

**Figure S4:** Satellite image of "Village 1" (1000ft scale); blue markers indicate ARISense monitoring sites (ARI014), red crosshatches indicate nearby biomass cookstoves. In addition to the explicitly marked cookstoves, each white square is a household, likely with its own cookstove. ARI014 was deployed to Village 1 site and was mounted on the roof of the residence of the village chief (4 m above ground) in the Makaula village of Mulanje, Malawi for 384 days from 11 July 2018 to 30 July 2018. Image source: Google Earth Pro Version 7.3.4.8248. *Makaula village, Mulanje, Malawi.* 16.045°S, 35.555°E, eye elevation 645 m. Borders and labels layer. Accessed: June 5, 2020.

The conclusions could be tightened up. What are the absolute main points your study, in particular, found? The novel work here, as stated at least in the introduction and abstract, is the comparison of different correction models over time. This gets a bit lost in the weeds.

> ➢ **We thank the reviewer for pointing out that we missed an opportunity to reinforce the main messages of our paper. We restructured and rewrote the conclusions in a way that better connects the objectives to our main findings and takeaways (shown in bolded text):**
>

[revised manuscript text omitted]

---

## Author Comment (AC2)

We would like to thank both reviewers for their helpful feedback. The revisions involve clarifications to the text, reordering of the sections to improve flow, and changes to the way the data are presented in Figures 1, 2 and 6.

Below we respond to the reviewer #2's comments in detail. Reviewer comments are in black text while author responses are in **bold purple text.** Excerpts from the original manuscript are shown in *italicized purple text.* Revisions to the manuscript are shown in *italicized red text.*

Note that reviewer #2 commented directly on a PDF version of the manuscript. To respond to these specific comments, we excised them and listed them as bullet points under the corresponding Line number.
* * *
Improved semi-conductor technology has made possible the significant evolution witnessed around the development and use of low cost air quality sensor in the last couple of years. The increased use of low-cost air quality (LCAQ) sensors in Africa, especially Sub-Saharan Africa has been brought about by its affordability and relative ease of deployment. However, the accuracy and quality of the measurement have been questionable and hugely debated in the scientific community. If the quality of the data from the sensors are improved, LCAQ sensors would bring a great revolution to air quality monitoring globally and enhance the our understanding of the problem, especially in low and middle income countries (LMICs) where the problem of air quality is endemic but reference grade instrument are not available.

**General Comments**

The study discusses an approach to enhance the quality of data obtained from units of LCAQ sensors ($O_X$, NO, $NO_2$, CO and PM) in Malawi by calibration using pre- and post-deployment collocations and five model approaches. The structure and layout of the manuscript makes it very difficult to follow through and understand.

> **To improve readability, we reorganized the subsections in the Results to improve the flow:**
>> o **3.1 Gas sensor performance during colocation**
>> o **3.2 OPC-N2 performance during colocation**
>> o **3.3 Gas sensor performance during deployment**
>> o **3.4 OPC-N2 performance during deployment**
>> o **3.5 Comparison of ARISense CO to remote sensing data**
>> o **3.6 Comparison to other ambient measurements in SSA**
>> o **3.7 Performance of ARISense sensor packages over time**
> **Further, to clearly establish the goal and objective of this paper, we added the following statement in the Introduction:**
>
> "*Our overarching goal was to assess the viability of establishing and maintaining a small, temporary network of LCS monitors in Malawi until a more formal governmental regulatory monitoring system can be established. Given that comparison to regulatory grade equipment in Malawi was not possible, the objective of this work was to devise an*

*alternative methodology to evaluate the ARISense technology for accuracy, precision, and stability over the 1-year pilot deployment.***"**

➢ **We also added an overview statement of the paper's organization in the Introduction. The last paragraph of Section 1 was rewritten as:**

**"***In Section 2.3 and 2.4, we discuss separate collocations of the gas sensors (in North Carolina, USA) and particle sensor (in Mulanje, Malawi) with reference or semi-reference instruments (Section 2.2). We use the collocation data and quantitative assessment metrics (Section 2.5) to compare the performance of five modelling approaches to calibrate the gas sensors (Section 3.1) and estimate error in the particle sensor data (Section 3.2). After deployment to Malawi (Section 2.6), we qualitatively assess how the ARISense performed in the field using contextual information about nearby emission sources, diurnal trend data, and an inter-comparison of calibrated gas model observations (Section 3.3 and 3.4). In Section 3.5 and 3.6, we compare the results to remote sensing and reanalysis data products and surface measurements from similar environments in SSA. Finally, in Section 3.7, we qualitatively assess the long-term stability of the sensor readings and calibration models in Malawi by comparing seasonally similar ambient data collected 1-year apart at the same location. In concluding (Section 4), we use evidence from this pilot study to characterize the benefits, limitations, and robustness of this technology and methodology for our application: collecting AQ data in under-studied regions. Additionally, we offer guidance on considerations to improve future remote deployment efforts.***"**

The language also needs adequate tone up to enhance the flow.

➢ **We improved grammar and sentence structure throughout the paper. Importantly, we removed the excessive number of semicolons.**

The materials in the manuscript should be arranged as much as possible in the way they are referred to in the manuscript. This makes it easy for the reader to follow the manuscript and supplementary material together without having to flip through pages of the supplementary material haphazardly.

➢ **In the final manuscript, after revisions are finalized, we will reorder the the supplementary information to ensure that it chronologically aligns with references made in the main manuscript as best as possible.**

Some of the figure (I have indicated these in the annotated version of the manuscript) are difficult to understand in their present form.

➢ **We redesigned Figures 1, 2 and 6 to improve their clarity. Revisions to Figure 6 are shown in the supplement to this response. We invite Reviewer 2 to see the supplement attached to our response to Reviewer 1. That file contains revised versions of Figure 1 and 2.**

Some sub-sections could be further divided into sub-sub-section to improve the organization and readability of the manuscript. Overall, the manuscript needs serious reorganization, restructuring and editing to enhance its understanding.

> ➢ **We respond to the reviewer's specific comment about subdividing the sections in "Specific comments" below. We hope our above response detailing the reorganization of the sections addresses the remaining portion of this comment.**

**Specific comments**

I have included the specific comments in the annotated version of the manuscript.

Line 17: What are the basis for selecting these five models?

> ➢ **We added the following text to Section 2.3 to better motivate our selections:**
>
> *"The five models were selected based on their performance in previous studies. The kNN hybrid model was found to enable accurate measurements even when pollutant levels were higher than encountered during calibration (Hagan et al., 2018). Given that we expected pollution levels to be higher in Malawi than during calibration in N.C., we expected kNN hybrid models to perform well for our unique data set. Further, the authors indicated that the kNN hybrid approach was expected to be widely applicable to a range of pollutants, sensors, and environments (Hagan et al., 2018). In a calibration and validation study conducted by Malings et al. (2019a), RF hybrid models were recommended for any low-cost monitor using electrochemical sensors similar to their sensor package, the Real-time Affordable Multi-Pollutant (RAMP) monitor. Given that the RAMP and ARISense monitors use the same electrochemical sensors and have similar integrated designs, we expected RF hybrid models to perform well for our dataset. HDMR models were found to effectively model interference effects derived from the variable ambient gas concentration mix and changing environmental conditions over three seasons for the sensor types used in the ARISense package (Cross et al., 2017). Finally, MLR and QR are some of simplest and most popular calibration approaches and they were included in this study for that reason."*

Line 34-35: Sources of air pollution in SSA are expected to increase over time given the regional growth in population and energy demand, a biomass fuel dominated energy mix, and slash and burn agricultural practices.

> ➢ Source or level?
> ➢ ?????
> ➢ **This statement was rephrased to "***Air pollution in SSA is expected to increase over time given regional growth in population and energy demand combined with a biomass fuel dominated energy mix.***"**

Line 50: Given the potential applications, LCS deployments are increasingly common (Giordano et al., 2021).

> ➢ ???? do you mean becoming common?
> ➢ **This statement was rephrased to "***becoming common***".**

Line 96-97: The ARISense sensors were collocated with reference equipment in North Carolina (NC) before and after deployment to Malawi. One OPC-N2, the ARISense particle sensor, was collocated with a semi-reference instrument at a field site in Malawi.

- Is the OPC-N2 part of the ARISense sensor package or is it a separate unit ? From Section 2.1, I think is a part of the ARISense package. If so, reframe this sentence.
- **This statement was rephrased to "***The ARISense were collocated with reference instruments in North Carolina (NC) before and after deployment to Malawi. One ARISense was collocated with a semi-reference PM instrument at a field site in Malawi to assess the performance of the integrated OPC-N2.***"**

Line 126-127: Previous validation studies found the MicroPEM performed well across a wide range of ambient PM concentrations and the real-time nephelometer, after gravimetric correction, agreed with fixed-site reference monitors.

- This sentence is confusing in its present state
- **This statement was rephrased to "***In previous evaluation studies, after gravimetric correction, the MicroPEM real-time nephelometer agreed with fixed-site reference monitors across a wide range of ambient PM concentrations (Du et al., 2019; Williams et al., 2014a).***"**

Line 132: 2.3 Gas sensor colocation and calibration

- There should be a subsection here that presents the component of the sensor units and the source of power. It should include a picture showing parts of the unit.
- **Information on the sensors and power source is currently presented in "Section 2.1 ARISense sensor packages". External and internal photographs of the ARISense are given in Sect 1. of Supplementary Information.**

Line 174: 2.5 Assessment metrics

- There should be a sub-section after this to discuss each of these metrics.
- **Quantitative descriptions for each metric are given in Sect 4. of the Supplementary Information. To make this easier for the reader to find, this statement: "***Quantitative descriptions for each metric are given in Sect. 4 of the Supplementary Information.***" was added immediately following the mention of the assessment metrics in Section 2.5.**

Line 177-178: Instead of the EPA recommended Root Mean Square Error (RMSE) metric, the Mean Absolute Error (MAE) was used to assess error in the estimated measurements compared to the true values.

- Is there any reason for opting for MAE instead of RMSE proposed by US EPA
- **We originally preferred MAE to RMSE as MAE does not weight outlying points as heavily as RMSE. However, after considering the reviewer's comment, we believe that adhering exactly to the EPA recommended metrics is a better approach. Therefore, we began replacing MAE with RMSE throughout the paper, and plan to only use RMSE in the revised and final manuscript. In our data sets, RMSE is generally <10% different from MAE, and this replacement does not change the interpretation of main results or the conclusions of the paper.**

Line 191: Supplementary Information, Sect. 5.

- The Google map in Figure S10 should be in the manuscript.
- **We agree with both reviewers who requested a map in the main manuscript. We plan to add a new figure to main paper (which would be Figure #1 in the revised**

**manuscript) showing a map of Malawi, denoting the monitoring sites, and a timeline of the project. A mock-up of this figure, based on Fig. S10, is shown in the supplement to our response to Reviewer #1.**

Line 212: RF hybrid model

➤ The authors need to present and discuss in details in section 2 the five models used in the calibration. You are presenting the results from the models without proper introduction to the models themselves.

➤ **We agree with the reviewer. We hope we resolved this issue with our response to the first specific comment above.**

Line 213: sensors

➤ The authors need to reframe this paragraph and find a way to get rid of the semi-colons.

➤ **This section was revised to remove semi-colons.**

Line 214: similarly

➤ Similarly? How do you mean?

➤ **"***performed similarly***" was replaced by "***returned similar performance metrics***"**

Line 214: MA

➤ What is MAE? Discuss the statistical tools used for your analyses before presenting the results.

➤ **Mean absolute error (MAE) is introduced as a statistical tool in "Section 2.5 Assessment Metrics". The formula is given in Sect. 4 of the Supplementary Information. Note that we will update the formulas to include Root Mean Square Error (RMSE) in the final manuscript.**

Line 215: The NO and CO sensors performed similarly, considering MAE values compared to the typical ambient concentration ranges; ambient CO concentrations are generally 1-2 orders of magnitude larger than NOx.

➤ This sentence is confusing. Be explicit.

➤ **This sentence was rephrased to "***The NO and CO sensors returned similar performance metrics, considering ambient CO concentrations are generally 1-2 orders of magnitude higher than NOx concentrations. The CO RMSE values (40-70 ppb) were correspondingly 1 order of magnitude larger than NOx RMSE values (2-7 ppb).***"**

Line 235: 3.2 Gas sensor performance during deployment

➤ It should be tidier and more readable if you could breakdown the results and discussion for CO, NOx, Ox into 3.2.1, 3.2.2, and 3.2.3.

➤ **We intend to divide this section (which will be Section 3.3 in the revised manuscript) into two sub-sub sections (***3.3.1 Bivariate histograms*** and ***3.3.2 Diurnal trends***). We believe that if we divide these sub-sections further by gas type, there will be too many sub-sections. We believe we cannot divide only by gas type, given that the discussion in this section often compares across gas sensor types.**

Line 246: ;

- ➤ ????
- ➤ **Extraneous use of semicolons was addressed throughout the paper.**

Line 221: $O_3$, Line 282: Ox, & Line 335: O3

- ➤ Have you resolved OX to get O3? Are you using Ox and O3 interchangeably? Or did you estimate O3 for Ox? O3 or Ox?
- ➤ **We do not intend to use 'Ox' and 'O3' interchangeably. 'Ox' is used when referencing the sensor itself or the raw data voltage readings. 'O3' is used when referencing the calibrated Ox measurements. All references to "'O3' indicate a data+model product. All instances of O3 and Ox were corrected throughout the paper to match this definition.**
- ➤ **Further, the following clarifying statement was added to Sect. 2.3: "***Note that references to 'O3' indicate estimates made from calibrating the Ox data. References to 'Ox' indicate raw voltage measurements from the total oxidant sensor. 'Ozone' is used when referring to the gaseous air pollutant.***"**

Line 291: annual mean diurnal trends

- ➤ A comparison of the observation with models on season basis could produce better results. Malawi has two main seasons -  the cool dry season between May and October with mean temperatures of around 13C and the hot season between November and April with temperatures between 30-35 C.
- ➤ **We agree with the reviewer, and we note that the diurnal trends do vary by season. However, we think this is beyond the scope of this specific paper, as we are only using the diurnal trend to assess coherency between the models. We postulate that we would come to the same conclusion (that the kNN hybrid and RF hybrid perform the best) even if we looked at the seasonal differences. Further, since we do not plan to use different models for different seasons (we will select one model to apply to all our field data), we believe that evaluation of annual trends is sufficient for this analysis.**
- ➤ **We do plan to explore the impact of seasons on our findings in the second paper of this two-paper series.**

Line 339: LT

- ➤ What is LT?
- ➤ **'***LT***' was changed to "***local time***"**

Line 381: higher

- ➤ Slightly higher
- ➤ **'***higher***' was changed to '***slightly higher***'**

Line 394: was highest during the burning season (Figure 4).

- ➤ This is only obvious for the university site and not the village site. I suggest you include "especially at the university site".
- ➤ **"***especially at the university site***" was added to the existing sentence.**

Line 406: Total Column CO

> - Can you also use total column CO? Just in case you will get better agreement.
> - **Technically, we could use Total Column CO for comparison, but the surface CO mixing ratio value is most closely comparable to our ARISense surface observations. Both quantities are mixing ratios with the same units (ppb), allowing us to perform qualitative and quantitative comparisons. If we used the Total Column CO, we would be limited to doing a qualitative comparison (such as correlation).**
> - **Since we are not aiming to assess the quality of the MOPITT surface CO product in Africa, and only using it as a benchmark to compare the trends in our surface observations with, we think the surface CO product is the best choice.**
> - **Further, in absence of in-situ data, air quality managers would likely first use the surface mixing ratio satellite product to inform on-the-ground conditions, rather than the Total Column, so this makes it the logical dataset to compare with.**

Line 410: Presently,

> - How do you mean by presently? Do you mean more recent studies?
> - **"*Presently,*" was changed to "*Existing*"**

Line 425: biased high compared

> - ????
> - **This statement was changed to "*ARI013 $PM_{2.5}$ mass concentration measurements were higher than measurements made by ARI014 and ARI015 (slope > 1), despite all ARISense being in the same location.*"**

Line 437: five suggested EPA target values (m, b, MAE)

> - A background for this should have been set in section 2.
> - **These metrics were introduced in Section 2.5 Assessment metrics. Their formulas and the recommended target values are given in Table S4 of Sect. 4 of the Supplementary Information.**

Line 481: Figure 6

> - This figure is very confusing in its present state. It is really difficult to interpret this figure and understand the message you intend to put across to the reader.
> - **We made a few changes to this figure (revised version below) to aid interpretation: 1) removed black borders on points and histogram blocks to eliminate an unnecessary visual barrier, 2) colored the text labels to facilitate matching between the histogram bin, its corresponding data point, and the text label (when applicable), 3) selected a sequential colorblind friendly palette, which creates a more natural relationship between color and increasing value range (i.e., on panel (c), redder colors indicate drier ambient conditions, bluer colors indicate wetter ambient conditions).Finally, we bolded the text labels, the panel labels and the figure text title. We would like to note that in the revised and final manuscript, we plan to use RMSE on the y-axis instead of MAE, to address reviewer #2's earlier comment. Preliminary analysis suggests the difference between the 2 metrics is**

**around 10%, and thus we expect the main findings of the paper to remain unchanged.**

[Figure]

**Figure 1 (revised):** Performance comparison of the RH-corrected Alphasense OPC-N2 compared to the MicroPEM under different environmental conditions: (a) wind direction, (b) ambient concentration, and (c) relative humidity during collocation at the Village 2 site in Mulanje, Malawi. An individual data point represents the paired metrics (MAE and $R^2$) for the OPC-N2 for a specific range of each condition. The histograms (inset) show the normalized frequency distributions for the ranges of each condition recorded during the collocation period. The colored markers in each panel correspond to the colored histogram bins. The metrics were calculated from 60-min averaged RH-corrected OPC-N2 $PM_{2.5}$ concentrations compared to the MicroPEM mass-corrected nephelometer. MAE is mean absolute error, assuming the MicroPEM concentrations as the true values; $R^2$ is the coefficient of determination. The lower left corner region of each panel indicates the highest performance based on these metrics.

**References**

Du, Y., Wang, Q., Sun, Q., Zhang, T., Li, T., and Yan, B.: Assessment of PM2.5 monitoring using MicroPEM: A validation study in a city with elevated PM2.5 levels, Ecotox. Environ. Safe., 171, 518–522, https://doi.org/10.1016/j.ecoenv.2019.01.002, 2019.

Cross, E. S., Williams, L. R., Lewis, D. K., Magoon, G. R., Onasch, T. B., Kaminsky, M. L., Worsnop, D. R., and Jayne, J. T.: Use of electrochemical sensors for measurement of air pollution: correcting interference response and validating measurements, Atmos. Meas. Tech., 10, 3575–3588, https://doi.org/10.5194/amt-10-3575-2017, 2017.

Hagan, D. H., Isaacman-VanWertz, G., Franklin, J. P., Wallace, L. M. M., Kocar, B. D., Heald, C. L., and Kroll, J. H.: Calibration and assessment of electrochemical air quality sensors by co-location with regulatory-grade instruments, Atmos. Meas. Tech., 11, 315–328, https://doi.org/10.5194/amt-11-315-2018, 2018.

Malings, C., Tanzer, R., Hauryliuk, A., Kumar, S. P. N., Zimmerman, N., Kara, L. B., Presto, A. A., and R. Subramanian: Development of a general calibration model and long-term performance evaluation of low-cost sensors for air pollutant gas monitoring, Atmos. Meas. Tech., 12, 903–920, https://doi.org/10.5194/amt-12-903-2019, 2019a.

Williams, R., Kaufman, A., Hanley, T., Rice, J., and Garvey, S.: Evaluation of Field-deployed Low Cost PM Sensors, U.S. Environmental Protection Agency, Washington, DC, 2014a.